# SimPlex-GT: A Simple Node-to-Cluster Graph Transformer for synergizing homophily and heterophily in Complex Graphs

## Abstract

Graph neural networks (GNNs) have proven effective on homophilic graphs, where connected nodes share similar features. However, real-world graphs often exhibit mixed patterns including heterophily, where connected nodes differ significantly. Traditional GNNs struggle with such cases due to their inherent smoothing operations. To address this limitation, we propose **SimPlex-GT**, a novel Graph Transformer (GT) model that synergizes homophilic and heterophilic patterns by integrating local GNN message passing with a novel global node-to-cluster (N2C) attention mechanism. Our approach disentangles node representations into local and global components: local features model neighborhood similarity, while global features attend to dynamic cluster prototypes learned on the fly. A learnable gating mechanism fuses these complementary views, and an orthogonality constraint encourages representational diversity. SimPlex-GT is trained under a self-supervised teacher–student architecture where the teacher sees the full graph and the student learns from masked inputs, with alignment enforced in a joint latent space. A dynamic masking strategy further emphasizes difficult nodes, based on prediction discrepancies. Comprehensive theoretical analysis demonstrates its strong capability, and extensive evaluations across 11 benchmark datasets show that SimPlex-GT achieves state-of-the-art performance on heterophilic graphs and remains highly competitive on homophilic graphs, all with superior computational efficiency. Code will be released upon acceptance.

## 1 Introduction

Graph-structured data are ubiquitous, naturally arising in social networks, knowledge graphs, communication networks, and beyond. Learning meaningful representations from graph-structured data has become an active and influential research direction, underpinning a variety of fundamental graph learning problems such as node classification, link prediction, and graph-level classification (Kipf & Welling, 2016; Gasteiger et al., 2019; Veličković et al., 2017; Wu et al., 2019). These problems arise across numerous application domains, including recommender systems, biological networks, and transportation infrastructures (Tang et al., 2020; Sankar et al., 2021; Fout et al., 2017; Wu et al., 2022b; Zhang et al., 2024). Among existing approaches, Graph Neural Networks (GNNs) have emerged as the prevailing framework, providing expressive architectures for capturing both local level patterns and global level structure (Hamilton, 2020; Gasteiger et al., 2018; Veličković et al., 2017; Rampášek et al., 2022).

Although traditional GNNs have demonstrated strong potential on graph-structured data, they suffer from inherent limitations. To capture long-range dependencies, they typically rely on stacking multiple layers, which often leads to well-known issues such as over-smoothing and over-squashing. More critically, their core inductive bias—message passing restricted to local neighborhoods—is well-suited for homophilic graphs, where connected nodes tend to be similar, but it fundamentally limits their effectiveness on heterophilic graphs, where connected nodes are often dissimilar. Recently, Graph Transformers (GTs) (Dwivedi & Bresson, 2020; Rampášek et al., 2022) have emerged as a promising alternative, mitigating over-smoothing and over-squashing through global attention that enables interactions between all node pairs. Crucially, GTs can identify similarities among nodes that are not directly connected, making them particularly effective for heterophilic graphs where informative relationships extend beyond local neighborhoods. Empirical studies further demonstrate

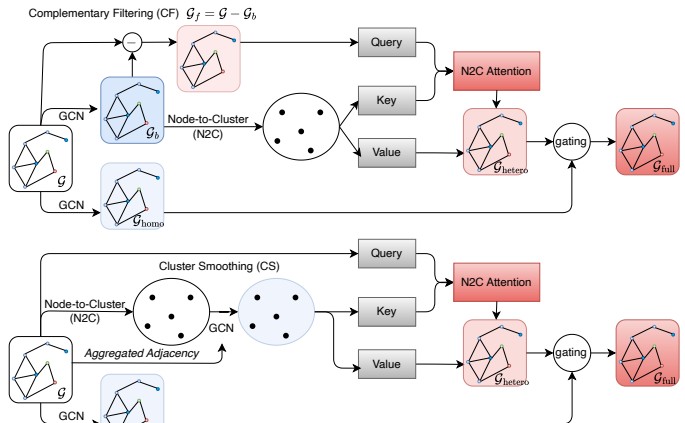

Figure 1: Illustration of our proposed method of synergizing homophily and heterophily in graph representation learning. We harness traditional GCN for addressing homophily and our proposed node-to-cluster attention for handling heterophily. We present two methods for the synergy. *Top:* Complementary Filtering (CF) based synergy, and *Bottom:* Cluster Smoothing (CS) based synergy. See text for details.

that GTs achieve strong performance on various graph learning tasks, underscoring their potential as a versatile architecture for diverse graph learning scenarios.

However, Graph Transformers (GTs) have notable limitations. First, their uniform global attention lacks the local inductive bias crucial for homophilic graphs, where nearby nodes with similar features provide the most relevant signals—something GNNs naturally capture through message passing. This makes GTs less effective on graphs with strong local structure or mixed homophily and heterophily, as neither GTs nor GNNs alone can model such complexity well. Second, GTs incur high computational costs due to their quadratic complexity $\mathcal{O}(N^2)$, limiting scalability. While some works add GNN branches to GTs for local context, this often leads to redundancy, training instability, and worse generalization on label-scarce datasets, while further increasing the computational burden. We provide a thorough review of related work in Appendix A.

To address those challenges, we propose a novel Graph Transformer architecture (Fig. 1) that synergizes homophily and heterophily existing in complex graphs, enabling the model to effectively handle complex structural patterns. We first propose a node-to-cluster (N2C) attention mechanism, a highly efficient representatinally-dense-computationally-sparse GTs with near-linear complexity that achieves SOTA performance on heterophilic graphs by its own. However, due to the lack of graph structural awareness, such models underperform on homophilic graphs. To overcome this issue, we introduce two novel designs to empower the N2C attention—complementary filtering (CF) and cluster smoothing (CS) with an orthogonality regularization term and a lightweight GCN residual pathway—which effectively fuse local and global information while providing enhanced training stability, as theoretically proven under common assumptions. Our model is termed **SimPlex-GT**, **Sim**plifying the com**plex**ity of mixed patterns in graph representation learning with GT.

Since node features play a crucial role in mixed-pattern graphs and limited labels in common semi-supervised setting often cause overfitting, we adopt an end-to-end self-supervised learning paradigm. Specifically, we design a feature alignment task in a joint latent space with a teacher–student architecture: the teacher processes the full graph while the student observes a partially masked graph. This approach removes the need for complex negative sampling in contrastive learning and avoids the space mismatch problem in generative learning. Finally, we propose an adaptive masking strategy that dynamically selects difficult nodes, encouraging the model to focus on the most challenging parts of the graph and thereby improving representation quality across diverse structural patterns.

**Our Contributions.** We make three main contributions: (i) **Unified Modeling of Homophily and Heterophily:** We propose **SimPlex-GT**, a novel Graph Transformer framework that effectively handles both homophilic and heterophilic structures by synergistically combining local GCN-based message passing with global node-to-cluster (N2C) attention. (ii) **Node-to-Cluster Attention Mechanism:** We introduce a sparse and learnable node-to-cluster attention module that replaces costly node-to-node attention with efficient and scalable interactions through dynamic cluster prototypes. This enables linear time complexity and improved robustness to structural noise in heterophilic graphs. (iii) **Comprehensive Theoretical Analysis and Strong Empirical Performance with Efficiency:** SimPlex-GT achieves state-of-the-art accuracy on heterophilic graphs and remains highly competitive on homophilic graphs across 11 benchmark datasets, while offering superior memory and training efficiency compared to prior Graph Transformers and specialized GNNs.

## 2 METHOD

In this section, we first show Graph Transformers (GTs) are strong learners for addressing heterophily in graph data, but GTs suffers from the quadratic complexity due to the node-to-node attention (Section 2.1.1). We then address this issue by proposing node-to-cluster attention with an end-to-end learnable clustering module (Section 2.1.2). However, this comes at the cost of sacrificing structural awareness, which leads to degraded performance on homophilic graphs. We handle this limitation by joint-forcing the node-to-cluster attention and GCN to harness the best of the two models to synergize homophily and heterophily (Section 2.2). Our final model is an integrative two-branch lightweight network. We train it under a self-supervised learning setting (Section 2.3).

### 2.1 GRAPH TRANSFORMERS ARE STRONG LEARNERS FOR HETEROPHILIC GRAPHS

Consider a graph $\mathcal{G} = <\mathcal{V}, \mathcal{E}>$ with a node set $\mathcal{V}$ and an edge set $\mathcal{E}$. For each node $v \in \mathcal{V}$, we have a node feature $f(v)$ and a node label $y_v$. The homophily ratio of $\mathcal{G}$ is defined by $\rho = \frac{\left|\{(u,v)\in\mathcal{E}|y_u=y_v\}\right|}{|\mathcal{E}|}$. Typically, homophilic graphs have homophily ratios closer to 1, while heterophilic graphs show homophily ratios closer to 0.

#### 2.1.1 PRELIMINARY ANALYSIS

As introduced in Sec. 1, GTs leverage global self-attention to enable nodes to aggregate information from semantically relevant but non-adjacent nodes, a property particularly beneficial for heterophilic graphs. To further clarify this advantage, we provide both empirical and theoretical analyses.

We first evaluate three GNN methods under supervised learning settings—GCN, GAT, and a vanilla GT with node-to-node attention—on four heterophilic graphs: Cornell, Texas, Wisconsin, and Actor, whose homophily ratios are 0.30, 0.11, 0.21, and 0.22, respectively.

As expected, we observe that vanilla GT significantly outperform classical GNNs on heterophilic graphs, particularly when the homophily ratio is low. Next, we provide a theoretical explanation of this phenomenon to support our claim that GTs are strong learners on heterophilic graphs.

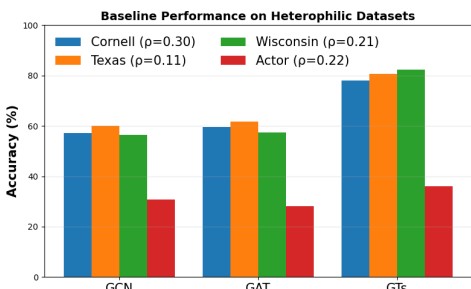

Figure 2: Performance on heterophilic graphs

**Theorem 1.** *Assume features align with labels in graphs:* $x^{(0)}(i) = y_i\mathbf{e}$ *with* $\|\mathbf{e}\| = 1$. *Let node-wise* $\rho_i := \frac{|\{j\in\mathcal{N}(i):y_j=y_i\}|}{|\mathcal{N}(i)|}$, $\mathcal{V}_{\text{hetero}} = \{i : \rho_i < \frac{1}{2}, i \in \mathcal{V}\}$, $\mathbf{h}^{(1)}(i)$ *is the node feature computed by a model (GNN or GT).*

- **GNN.** *For a one-layer mean-aggregation (linearized) GNN without self-loop,* $\langle\mathbf{h}^{(1)}(i), y_i\mathbf{e}\rangle = 2\rho_i - 1 < 0$ *for all* $i \in \mathcal{V}_{\text{hetero}}$, *so it misclassifies at least* $|\mathcal{V}_{\text{hetero}}|/|\mathcal{V}|$ *nodes.*
- **GT.** *There exist parameters* $\boldsymbol{\theta}^*$ *such that:* $\langle\mathbf{h}^{(1)}_{\boldsymbol{\theta}^*}(i), y_i\mathbf{e}\rangle > 0, \forall i \in \mathcal{V}$, *yielding zero error.*

The proof is provided in Appendix B. This theorem demonstrates that GT's capacity to learn global patterns enables it to rely primarily on feature similarity, thereby overcoming the structural constraints that lead to GNN failures in heterophilic settings. The key insight is that GTs can adaptively disregard the original graph topology when it is uninformative for the task (e.g., node classification).

However, GTs face two major challenges. *First*, their quadratic complexity of $O(N^2)$, where $N$ is the number of nodes, makes it impractical to apply full node-to-node attention on large-scale graphs (see the out-of-memory in Table 1). *Second*, node-to-node attention is an interaction-before-aggregation computational mechanism, making the resulted node representation at each layer less meaningful if there are hierarchical structures among nodes (which indeed often exist in real-world graph data).

#### 2.1.2 OUR PROPOSED NODE-TO-CLUSTER ATTENTION

We rethink node-to-node (N2N) attention by reversing its interaction-before-aggregation mechanism. We seek aggregation-before-interaction computing mechanisms. To be efficient and effective, we resort to node clustering as the aggregation strategy to exploit hierarchical node structures. To facilitate end-to-end training, we propose to directly learn a node-to-cluster module as to enable

joint clustering-for-meaningful-attention and attention-for-compact-clustering in the resulted node-to-cluster (N2C) attention model.

Following the terminology used in Transformers (Vaswani et al., 2017), denote by $X_{N,d}$ a sequence of tokens in the $d$-dim embedding space, extracted from a graph, where $N$ is the number of nodes, and $d$ the node feature dimensions. The core of node-to-node attention is,

$$\text{N2N Attention: } X'_{N,d} = \text{Softmax}\big(\frac{Q \cdot K^\top}{\sqrt{d}}\big)_{N \times N} \cdot V, \tag{1}$$

$$Q = X_{N,d} \cdot W^{d \times d}_{\text{query}}, \quad K = X_{N,d} \cdot W^{d \times d}_{\text{key}}, \quad V = X_{N,d} \cdot W^{d \times d}_{\text{value}}, \tag{2}$$

where we omit the (optional) bias terms in the linear transformations of computing the query, key and value for notation simplicity. The quadratic complexity lies in $QK^\top$, which is of $N \times N$ size.

In our proposed node-to-cluster attention, we keep the query unchanged, while computing the key and value not directly from $X_{N,d}$, but from a small predefined number $M \ll N$ of learned cluster tokens, $Z_{M,d}$ (e.g., $M = 5$). More specifically, we have,

$$Z_{M,d} = C^\top_{N,M} \cdot X_{N,d}, \tag{3}$$

$$C_{N,M} = \text{Softmax}_{\text{along } N}\big(X_{N,d} \cdot W^{d \times M}_{\text{cluster}}\big), \tag{4}$$

$$\mathbf{K} = Z_{M,d} \cdot W^{d \times d}_{\text{key}}, \quad \mathbf{V} = Z_{M,d} \cdot W^{d \times d}_{\text{value}}, \tag{5}$$

where $C_{N,M}$ is the N2C probabilistic assignment learned via a simple linear transformation, $W^{d \times M}_{\text{cluster}}$. Each of the cluster token $Z_m$ is thus a $C_{N,m}$ weighted sum of node tokens with the sum of weights equal to 1 ($\sum_{n=1}^N C_{n,m} = 1$) due to the Softmax in Eqn. 4. Eqn. 1 can be rewritten as,

$$\text{N2C Attention: } \quad X'_{N,d} = \text{Softmax}\big(\frac{Q \cdot \mathbf{K}^\top}{\sqrt{d}}\big)_{N \times M} \cdot \mathbf{V}, \tag{6}$$

which has linear complexity with respect to the number of nodes $N$.

In practice, we often use multi-head attention with $Q_{H,N,c} = \text{Reshape}(Q_{N,d})$, $\mathbf{K}_{H,M,c} = \text{Reshape}(\mathbf{K}_{M,d})$ and $\mathbf{V}_{H,M,c} = \text{Reshape}(\mathbf{V}_{M,d})$, where $H$ is the predefined number of heads, and $d = H \times c$. We have,

$$X'_{N,d} = \text{Reshape}\big(X'_{H,N,c}\big), \quad X'_{h,N,c} = \text{Softmax}\big(\frac{Q_{h,N,c} \cdot \mathbf{K}^\top_{h,M,c}}{\sqrt{c}}\big)_{N \times M} \cdot \mathbf{V}_{h,N,c}. \tag{7}$$

We use the proposed multi-head N2C attention in a GT block with post-norm in our experiments.

**Noise Reduction** Besides the advantage of linear complexity, our node-to-cluster attention allows each node to selectively align to the most relevant cluster prototype learned on the fly. This enables nodes to bypass local noise from inconsistent neighbors and directly attend to clusters that share feature-level consistency. We provide theoretical analysis in Theorem 2 with proof in Appendix C:

**Theorem 2** (Variance reduction: N2C vs N2N). *Assume $x_i^{(0)} = y_i \mathbf{e} + \varepsilon_i$ with $\mathbb{E}[\varepsilon_i] = 0$ and $\text{Cov}(\varepsilon_i) = \Sigma \preceq \sigma^2 I$. Let $p_{i,c} \in [0,1]$ with $\sum_{c=1}^C p_{i,c} = 1$ be soft cluster assignments, and cluster prototypes $\mathbf{p}_c = \frac{\sum_{i=1}^N p_{i,c} x_i^{(0)}}{\sum_{i=1}^N p_{i,c}}$. For any nonnegative attention weights summing to 1,:*

$$\text{Var}\Big(\sum_j \alpha_{i \to j} \langle x_j^{(0)}, \mathbf{e} \rangle\Big) \leq \sigma^2 \|\mathbf{e}\|^2, \qquad \text{Var}\Big(\sum_c \alpha_{i \to c} \langle \mathbf{p}_c, \mathbf{e} \rangle\Big) \leq \sigma^2 \|\mathbf{e}\|^2 \sum_c \alpha_{i \to c} r_c, \tag{8}$$

*where $r_c = \frac{\sum_i p_{i,c}^2}{(\sum_i p_{i,c})^2} \leq 1$ and $\sum_c \alpha_{i \to c} r_c \leq 1$. The N2C bound doesn't exceed the bound of N2N.*

**Remark 1.** *The variance reduction factor $r_c \approx 1/n_{eff}$ where $n_{eff}$ is the effective number of nodes in cluster c. Thus N2C provides denoising proportional to cluster sizes, while N2N lacks such benefit.*

### 2.1.3 IS GRAPH TRANSFORMER ENOUGH FOR HOMOPHILIY?

Both N2N and N2C GTs ignore structural information and treat all nodes equally during aggregation. As a result, their performance on homophilic graphs remains unclear, since structural signals play a much more important role in such settings. To investigate this, we conducted a preliminary experiment to evaluate both N2N and N2C attention on different type of graphs. Specifically, we use **four heterophilic** datasets, Cornell, Texas, Wisconsin and Actor, and **three homophilic** datasets, Cora, Citeseer, and Pubmed under a supervised learning setup. The results are shown in Table 1.

Table 1: Node classification accuracy (in percent $\pm$ standard deviation across 10 splits) under Supervised Learning (SL). OOM stands for "Out of Memory".

| Methods | Heterophilic | | | | Homophilic | | |
|---|---|---|---|---|---|---|---|
| | Cornell | Texas | Wisconsin | Actor | Cora | Citeseer | Pubmed |
| GCN | 57.03±3.30 | 60.00±4.80 | 56.47±6.55 | 30.83±0.77 | 81.50±0.30 | 70.30±0.27 | 79.00±0.05 |
| GAT | 59.46±3.63 | 61.62±3.78 | 54.71±6.87 | 28.06±1.48 | 83.02±0.19 | 72.51±0.22 | 79.87±0.03 |
| MLP | 81.08±7.93 | 81.62±5.51 | 84.31±3.40 | 35.66±0.94 | 56.11±0.34 | 56.91±0.42 | 71.35±0.05 |
| N2N GT (SL) | 81.35±2.55 | 84.59±4.20 | 87.43 ±3.21 | OOM | 66.80±0.12 | 65.20±0.32 | OOM |
| N2C GT (SL) | 82.16±1.79 | 87.60±4.26 | 88.23±3.18 | 36.13±0.62 | 73.16±0.18 | 68.12±0.30 | 74.90±0.08 |

From the results, we observe that our proposed N2C attention demonstrates clear advantages on heterophilic graphs, outperforming classical GNNs. Our N2C attention slightly outperforms N2N attention on those heterophilic graphs, and can handle larger graphs for which N2N attention suffers from OOM. However, **both N2N and N2C attention substantially underperform on homophilic graphs,** as they completely ignores the important structural information, including the clustering process in our N2C attention, which is consistent with our earlier claim. As highlighted in prior literature (Rampášek et al., 2022), GNNs remain necessary for effectively capturing local structural information, particularly when a graph exhibits mixed patterns of homophily and heterophily.

## 2.2 SYNERGIZING HOMOPHILY AND HETEROPHILY

As illustrated in Fig. 1, we present two novel designs: *Complementary Filtering (CF)* and *Cluster Smoothing (CS)*, by inducing graph structural awareness in either the input (via CF) or output (via CS) of the node-to-cluster module.

**Complementary Filtering (CF).** In the graph spectral domain, low-frequency components correspond to signals that vary slowly across neighboring nodes. In other words, the low-frequency spectrum captures the overall trends shared within the same community or class of nodes. Such patterns are smooth and noise-resistant, thus providing stable representations of node groups. Consequently, low-frequency information is well suited to serve as prototypes (i.e., cluster centers). In contrast, high-frequency components correspond to signals that exhibit sharp variations across neighboring nodes in the graph spectral domain. These variations makes high-frequency information highly discriminative. Therefore, high-frequency signals naturally serve as effective queries: they emphasize the distinctive aspects of each node that guide the selection of the most relevant prototype.

To formalize this idea, we are motivated by complementary filtering and decompose the node features $f(v)$ into two complementary channels via a low-pass filter $\mathcal{F}_{\text{lowpass}}$ which is implemented and approximated by a single layer of GCN for simplicity:

$$\mathcal{G}_b = \mathcal{F}_{\text{lowpass}}(\mathcal{G}), \qquad \mathcal{G}_f = \mathcal{G} - \mathcal{G}_b. \qquad (9)$$

As illustrated in Fig. 1 (top), we compute Query using $\mathcal{G}_f$, and apply the node-to-cluster module using $\mathcal{G}_b$. We provide a Theorem 3 on the stability of our CF design with proof in Appendix D.

**Theorem 3** (Stability). *Let queries use high-pass features and keys/values use low-pass prototypes:*

$$s_{i,c} = \frac{1}{\sqrt{d_k}}\langle Q(x_f^{(i)}), K(c_b^{(c)})\rangle, \quad s_{i,c}^{\star} = \frac{1}{\sqrt{d_k}}\langle Q(h^{(i)}), K(s^{(c)})\rangle. \qquad (10)$$

*Assume $Q, K$ are $L_Q, L_K$–Lipschitz and $\mathbb{E}\|K(c_b^{(c)})\|^2 \leq M_K^2$, $\mathbb{E}\|Q(h^{(i)})\|^2 \leq M_Q^2$. Then*

$$\mathbb{E}\big[(s_{i,c} - s_{i,c}^{\star})^2\big] \leq \frac{2}{d_k}\Big(L_Q^2 M_K^2\, \mathbb{E}\|x_f^{(i)} - h^{(i)}\|^2 + L_K^2 M_Q^2\, \mathbb{E}\|c_b^{(c)} - s^{(c)}\|^2\Big). \qquad (11)$$

*Consequently, this yields strictly smaller logit MSE compared to using unsplit features whenever the filters attenuate off-target spectra.*

**Cluster Smoothing (CS).** Alternatively, we can compute Query and apply the node-to-cluster module, both from the input graph, as illustrated in Fig. 1 (bottom). After obtaining clusters (Eqns. 3 and 4), similar nodes are grouped according to their raw features. These grouped nodes can be viewed as forming a new *coarse graph*, where each node corresponds to a cluster prototype. To incorporate structural information, instead of performing message passing over all nodes in the original graph, we smooth only the $M$ cluster prototypes on a coarse graph induced by the original topology. Denote by $A_{N \times N}$ the adjacency matrix of the input graph. With the clustering assignment $C_{N,M}$ (Eqn. 4), we compute the aggregated adjacency matrix $A_c$ among clusters and then smooth the clusters $Z_{M,d}$ (Eqn. 3) using $\mathcal{F}_{\text{lowpass}}$ (e.g., 1-layer GCN) by,

$$A_c = C^{\top}AC, \qquad Z' = \mathcal{F}_{\text{lowpass}}(Z, \widehat{A}_c) \qquad (12)$$

where $\widehat{A}_c$ is the normalized adjacency matrix of coarse graph $G_c$. We present the following theorem:

**Theorem 4** (Variance reduction in CS). *Assume unsmoothed prototype $z_c = \mu_c + \varepsilon_c$ with $\mathbb{E}[\varepsilon_c] = 0$, $\mathrm{Cov}(\varepsilon_c) \preceq \sigma^2 I$. Let $\mathcal{N}(c)$ denote neighbors of $c$ on the coarse graph $G_c$, $c \notin \mathcal{N}(c)$, $w_{cu} \geq 0$, and $\{\varepsilon_u\}$ independent across clusters. Let $\sum_{u \in \mathcal{N}(c)} w_{cu} = 1$, and define one-step residual smoothing: $\tilde{z}_c = (1 - \alpha) z_c + \alpha \sum_{u \in \mathcal{N}(c)} w_{cu} z_u$, where $\alpha \in (0, 1)$. Then for any unit direction $\mathbf{e}$,*

$$\mathrm{Var}\big(\langle \tilde{z}_c - \mu_c, \mathbf{e} \rangle\big) = \mathrm{Var}\Big(\big\langle (1-\alpha)\varepsilon_c + \alpha \sum_u w_{cu}\varepsilon_u, \, \mathbf{e} \big\rangle\Big) \leq \big((1-\alpha)^2 + \alpha^2\big)\sigma^2 \leq \sigma^2, \quad (13)$$

*with strict inequality whenever $0 < \alpha < 1$.*

See Appendix E for proof. It shows that a single low-pass smoothing reduces the directional variance of each cluster prototype. Thus, smoothed prototypes are uniformly more stable than unsmoothed ones, and the smoothing strength $\alpha$ provides a controllable variance–fidelity trade-off.

**Our Final Block: SimPlex-GT with Orthogonality Regularization.** To further induce graph structural awareness for addressing the homophily in graphs, we introduce a one-layer GCN branch as the residual pathway capturing those information, denoted by $\mathcal{G}_{\text{homo}}$, and integrate it to the heterophily-targeted N2C output $\mathcal{G}_{\text{hetero}}$. The final representation is,

$$\mathcal{G}_{\text{full}} = \mathcal{G}_{\text{homo}} + \mathcal{G}_{\text{hetero}}, \qquad \bar{\mathcal{G}}_{\text{full}} = \text{LN}\big(\mathcal{G}_{\text{full}} + \text{MLP}(\mathcal{G}_{\text{full}})\big). \qquad (14)$$

To prevent potential *representation redundancy and interference* in the simple sum of $\mathcal{G}_{\text{homo}}$ and $\mathcal{G}_{\text{hetero}}$, e.g., the two branches may collapse onto similar subspaces or create scale/gradient competition/confliction, diminishing complementary gains, we introduce an auxiliary *orthogonality regularization term* that encourages the two branches to learn features reinforcing each other. We minimize the node-wise cosine similarity between the two branches, which will be used as the auxiliary loss in training with a tunable weight $\lambda_{\text{orth}} > 0$. Note that, similar to other baselines (Rampášek et al., 2022), above designs can be viewed as a building block in our framework, and multiple blocks can be easily stacked to enhance the model's expressive power. In our experiments, we retain a single block for simplicity.

### 2.3 SELF-SUPERVISED LEARNING OFFERS A ROBUST LEARNING PARADIGM

To reveal the full potential of our proposed SimPlex-GT, we adopt a self-supervised learning framework, which yields more robust node representations by leveraging the synergistic two-branch modeling power. We exploit masked node modeling as the primary proxy objective and employ a teacher–student predictive architecture, as they have shown strong representational learning capability in domains such as computer vision (Assran et al., 2023; 2025).

Given that graph data possesses unique structural connections that distinguish it from other sensory data such as imagery, we enforce prediction between student and teacher models across all nodes, rather than limiting it to masked nodes only. This design acknowledges the dependencies between masked and unmasked nodes. Furthermore, to better tackle the mixed patterns in graph data, we propose a node-difficulty–driven dynamic masking strategy that adaptively adjusts the masking process, enabling the model to learn more robust and informative representations.

**Masked Node Modeling with Teacher–Student SSL.** Given a graph $\mathcal{G}$ and a node-wise mask $\mathcal{M}$, we divide the nodes into the masked set $\mathcal{V}_m = \{v \mid \mathcal{M}(v) = 1\}$ and the unmasked set $\mathcal{V}_u = \mathcal{V} \setminus \mathcal{V}_m$. For $v \in \mathcal{V}_m$, the raw features are replaced with random noise $f(v) \sim \mathcal{N}(0, 1)$, yielding the partially masked graph $\mathbb{G} = (\mathcal{V}_u \cup \mathcal{V}_m, E)$. We adopt a teacher–student predictive framework: the student $S(\cdot; \phi)$ sees $\mathbb{G}$, while the teacher $T(\cdot; \psi)$ sees the full graph $\mathcal{G}$. Both share the same SimPlex-GT, but the teacher is updated via exponential moving average (EMA) of the student parameters without requiring gradients to ensure training stability (as commonly done in masked data modeling).

**All-Node Latent Prediction.** Instead of predicting only masked nodes, we enforce consistency over the entire node set to capture local-global contextual adaptations. For each $v \in \mathcal{V}$, the outputs of student and teacher are $S(v; \phi), T(v; \psi) \in \mathbb{R}^d$, and the prediction loss is

$$\mathcal{L}(\phi) = \frac{1}{N} \sum_{v \in \mathcal{V}} \Big( \|S(v; \phi) - T(v; \psi)\|_2^2 + \lambda_{\text{orth}} \cdot \text{Cos}\big(f_{\text{homo}}^S(v), f_{\text{hetero}}^S(v)\big) \Big), \qquad (15)$$

where $\text{Cos}\big(f_{\text{homo}}^S(v), f_{\text{hetero}}^S(v)\big)$ is the auxiliary orthogonality regularization term using the cosine similarity between the two branches in the student network, and $\lambda_{\text{orth}}$ the trade-off parameter.

**Node-Difficulty–Driven Dynamic Masking.** Let $R \in (0, 1)$ be the overall masking ratio, so that $M = \lfloor N \cdot R \rfloor$ nodes are masked per iteration. Training begins with random masking (as warm-up)

Table 2: Node classification accuracy (%) reported as mean ± standard deviation across 10 data splits. The best, runner-up and third best results for each dataset are highlighted.

| Category | Methods | Heterophilic Datasets | | | | Homophilic Datasets | | | |
|---|---|---|---|---|---|---|---|---|---|
| | | Cornell | Texas | Wisconsin | Actor | Cora | CiteSeer | PubMed | Arxiv |
| | Homo Ratio | 0.30 | 0.11 | 0.21 | 0.22 | 0.81 | 0.74 | 0.80 | 0.66 |
| Semi-Supervised Learning (SL) | | | | | | | | | |
| GNNs | GCN | 57.03±3.30 | 60.00±4.80 | 56.47±6.55 | 30.83±0.77 | 81.50±0.30 | 70.30±0.27 | 79.00±0.05 | 71.74±0.27 |
| | GAT | 59.46±3.63 | 61.62±3.78 | 54.71±6.87 | 28.06±1.48 | 83.02±0.19 | 72.51±0.22 | 79.87±0.03 | 71.92±0.17 |
| | MLP | 81.08±7.93 | 81.62±5.51 | 84.31±3.40 | 35.66±0.94 | 56.11±0.34 | 56.91±0.42 | 71.35±0.05 | 55.50±0.23 |
| | WRGAT | 81.62±3.90 | 83.62±5.50 | 86.98±3.78 | 36.53±0.77 | 81.97±1.50 | 70.85±1.98 | 80.86±0.55 | — |
| | H2GCN | 82.16±4.80 | 84.86±6.77 | 86.67±4.69 | 35.86±1.03 | 81.76±1.55 | 70.53±2.01 | 80.26±0.56 | — |
| GTs | GraphGPS | 66.22±3.87 | 75.41±1.46 | 78.04±2.88 | 36.95±0.65 | 82.84±1.03 | 72.73±1.23 | 79.94±0.26 | 70.97±0.41 |
| | NAGphormer | 56.22±8.08 | 63.51±6.53 | 62.55±6.22 | 34.33±0.94 | 82.12±1.18 | 71.47±1.30 | 79.73±0.28 | 70.13±0.55 |
| | Exphormer | 54.05±4.41 | 77.84±2.21 | 69.94±3.33 | 35.77±0.45 | 82.77±1.38 | 71.63±1.19 | 79.46±0.35 | 72.44±0.28 |
| | ‡ NodeFormer | 65.77±4.59 | 69.37±2.55 | 73.86±4.33 | 35.11±1.13 | 82.20±0.90 | 72.50±1.10 | 79.90±1.00 | 59.90±0.42 |
| | ‡ SGFormer | 74.97±4.31 | 76.15±1.99 | 78.79±2.89 | 37.46±0.84 | 84.50±0.80 | 72.60±0.20 | 80.30±0.60 | 72.63±0.13 |
| Self-Supervised Learning (SSL) | | | | | | | | | |
| Homo | DGI | 63.35±4.61 | 60.59±7.56 | 55.41±5.96 | 29.82±0.69 | 82.29±0.56 | 71.49±0.14 | 77.43±0.84 | 70.19±0.73 |
| | MVGRL | 64.30±5.43 | 62.38±5.61 | 62.37±4.32 | 30.02±0.70 | 83.03±0.27 | 72.75±0.46 | 79.63±0.38 | 70.88±0.51 |
| | BGRL | 57.30±5.51 | 59.19±5.85 | 52.35±4.12 | 29.86±0.75 | 81.08±0.17 | 71.59±0.42 | 79.97±0.36 | 71.24±0.35 |
| | GRACE | 54.86±6.95 | 57.57±5.68 | 50.00±5.83 | 29.01±0.78 | 80.08±0.53 | 71.41±0.38 | 80.15±0.34 | 70.96±0.31 |
| | GraphMAE | 61.93±4.59 | 67.80±3.37 | 58.25±4.87 | 31.48±0.56 | 84.20±0.40 | 73.20±0.39 | 81.10±0.34 | 71.75±0.17 |
| Hetero | DSSL | 53.15±1.28 | 62.11±1.53 | 56.29±4.42 | 28.36±0.65 | 83.06±0.53 | 73.20±0.51 | 81.25±0.31 | 70.13±0.25 |
| | HGRL | 77.62±3.25 | 77.69±2.42 | 77.51±4.03 | 36.66±0.35 | 80.66±0.43 | 68.56±1.10 | 80.35±0.58 | 68.55±0.38 |
| | * S3GCL | 81.27±3.67 | 86.12±3.91 | 84.56±2.71 | 36.88±0.34 | *— | *— | *— | 71.36±0.60 |
| | †MUSE | 82.00±3.42 | 83.98±2.81 | 88.24±3.19 | 36.15±1.21 | 82.22±0.21 | 71.14±0.40 | 82.90±0.40 | 70.98±0.32 |
| | GREET | 73.51±3.15 | 83.80±2.91 | 82.94±5.69 | 35.79±1.04 | 83.84±0.71 | 73.25±1.14 | 80.29±1.00 | 71.09±0.43 |
| Ours | SimPlex-GT -CF | 83.78±4.68 | 92.97±4.39 | 91.37±2.35 | 37.51±0.87 | 84.05±0.15 | 73.28±0.12 | 81.25±0.26 | 72.01±0.26 |
| | SimPlex-GT -CS | 84.86±3.24 | 92.97±3.66 | 92.16±2.63 | 37.68±0.83 | 83.85±0.12 | 73.12±0.20 | 80.88±0.36 | 71.97±0.32 |

‡ Original papers report limited heterophilic results, we supplement them with our reproduction and those from (Tang et al., 2025).

† MUSE provides hyperparameters only for Cornell, with unreproducible results; we tuned the others ourselves.

⋆ As S3GCL's code is under construction, we report published results except on Cora, CiteSeer, and PubMed, which use different splits with higher label rates.

and then switches to an exploitation–exploration scheme: $m = \lfloor r \cdot M \rfloor$ nodes are sampled based on their prediction difficulty (exploitation), while the remaining $(M - m)$ are sampled randomly (exploration), where $r \in (0, 1)$ is the exploitation ratio, and the difficulty level of a node $v$ is simply defined by $\tau(v) = \|S(v; \phi) - T(v; \psi)\|_2^2$. Overall, each node $v$ is masked according to a Bernoulli distribution with probability,

$$p_v = p_0 + \delta_v, \quad p_0 = (1 - r)R, \quad \delta_v = \frac{\tau(v)}{\tau_{\max}} \cdot r \cdot R, \tag{16}$$

where $\tau_{\max} = \max_{u \in \mathcal{V}} \tau(u)$. Thus all nodes retain a base masking rate $p_0$ to prevent biased sampling from over-focusing on difficult nodes, while harder nodes are masked more frequently. This design balances task difficulty with data diversity, encouraging robust representation learning.

## 3 EXPERIMENTS

**Datasets.** We evaluate our model on a diverse suite of **11 real-world benchmarks**, comprising **four widely used homophilic graphs** (Cora, CiteSeer, PubMed, and ArXiv) (Sen et al., 2008; Hu et al., 2021) and **seven heterophilic graphs** (Cornell, Texas, Wisconsin, Actor, Chameleon, Squirrel, and Roman-Empire) (Pei et al., 2020; Platonov et al., 2023). These datasets cover both small- and large-scale networks and originate from different domains, ensuring a comprehensive evaluation. As the original Chameleon and Squirrel datasets are known to present issues (Platonov et al., 2023), we use their filtered versions to provide a more reliable assessment of model performance.

For clarity, we also report the homophily ratios of all datasets in the main results table, with full dataset statistics summarized in Appendix G. To ensure fair comparisons, we follow the standard data splits provided by either the original papers (Platonov et al., 2023) or PyG (Fey & Lenssen, 2019). Specifically, for heterophilic graphs we use all ten official splits, while for Cora, CiteSeer, and PubMed we adopt PyG's commonly used low-label split with 20 labeled nodes per class. For ArXiv, we use the official PyG split and report results averaged over 10 runs.

**Baselines.** To **ensure consistency** with prior work, we adopt the widely used node classification task as our primary downstream evaluation. We compare against two groups of baselines: (1) **semi-supervised learning methods**, including traditional GNNs such as GCN (Kipf & Welling, 2016), GAT (Veličković et al., 2017), and a simple MLP; heterophily-oriented GNNs such as

WRGAT (Suresh et al., 2021), H2GCN (Zhu et al., 2020a), GPR-GNN (Chien et al., 2020), and FAGCN (Bo et al., 2021); and Graph Transformers including GraphGPS (Rampášek et al., 2022), NAGphormer (Chen et al., 2022b), Exphormer (Shirzad et al., 2023), NodeFormer (Wu et al., 2022a), and SGFormer (Wu et al., 2023); and (2) **self-supervised learning methods**, including general SSL approaches such as DGI (Velickovic et al., 2019), MVGRL (Hassani & Khasahmadi, 2020), BGRL (Thakoor et al., 2021), GRACE (Zhu et al., 2020b), and GraphMAE (Hou et al., 2022); as well as SSL methods tailored for heterophilic graphs such as DSSL (Xiao et al., 2022), HGRL (Chen et al., 2022a), S3GCL (Wan et al., 2024), GREET (Liu et al., 2022), and MUSE (Yuan et al., 2023).

For evaluation, we adopt the commonly used protocol in SSL studies (Yuan et al., 2023), where the pretrained SSL model is frozen and student model's node embeddings are fed into a linear classifier for downstream node classification. To ensure fairness, we reproduce the results of representative baselines (Liu et al., 2022; Hou et al., 2022; Yuan et al., 2023; Wu et al., 2023; 2022a) using the hyperparameters released in their official implementations, while keeping the data splits identical across all methods. For the remaining baselines, we report numbers from their original publications or from other benchmark studies (Yuan et al., 2023; Xiao et al., 2024; Wan et al., 2024; Tang et al., 2025). A complete list of hyperparameter searching space is provided in Appendix H.

Table 3: Node classification accuracy (%) on large heterophilic graphs

| Methods | Chameleon(filtered) | Squirrel(filtered) | Roman-Empire |
|---------|---------------------|--------------------|--------------|
| Homo Ratio | 0.24 | 0.21 | 0.05 |
| **Semi-Supervised Learning (SL)** | | | |
| GCN | 40.89±4.12 | 39.47±1.47 | 73.69±0.74 |
| GPR-GNN | 39.93±3.30 | 38.95±1.99 | 64.85±0.27 |
| FAGCN | 41.90±2.72 | 41.08±2.27 | 65.22±0.56 |
| H2GCN | 26.75±3.64 | 35.10±1.15 | 60.11±0.52 |
| GraphGPS | 40.79±4.03 | 39.67±2.84 | 82.00±0.61 |
| NodeFormer | 34.73±4.14 | 38.52±1.57 | 64.49±0.73 |
| SGFormer | 44.93±3.91 | 41.80±2.27 | 79.10±0.32 |
| **Self-Supervised Learning (SSL)** | | | |
| DGI | 32.61±2.92 | 38.78±2.34 | 43.16±0.78 |
| BGRL | 32.55±4.65 | 35.67±1.42 | 52.16±0.25 |
| GRACE | 35.39±3.58 | 36.21±2.81 | 51.58±0.98 |
| MUSE | 46.48±2.51 | 41.57±1.44 | 66.26±0.53 |
| GREET | 44.67±2.98 | 39.69±1.85 | 63.37±1.91 |
| SimPlex-GT - CF | 50.32±3.20 | 47.75±1.65 | 81.56±0.68 |
| SimPlex-GT - CS | 50.48±3.22 | 47.56±1.63 | 79.88±0.78 |

## 3.1 NODE CLASSIFICATION RESULTS ANALYSIS

Tables 2 and 3 show the results for node classification, from which we draw the following conclusions:

- Our SimPlex-GT significantly outperforms almost all supervised (both GNNs and GTs) and self-supervised baselines across heterophilic graphs of varying scales, and achieves on-par performance with the SOTA baselines on homophilic graphs. For example, our method outperforms the state-of-the-art baselines by 6.85% on Texas, 4.00% on Chameleon, and 5.95% on Squirrel. These gains stem from the effectiveness of our node-to-cluster attention design (also see Table 1) and the synergy between local and global information.
- Regarding baselines, on heterophilic graphs, classical GNNs perform poorly, while heterophily-oriented GNNs (e.g., H2GCN, WRGAT, FAGCN) achieve substantial improvements. GTs also show competitive results on heterophilic datasets, even without explicit designs for heterophily. On homophilic graphs, both classical GNNs and GTs perform competitively and GTs often achieve better results.
- In terms of SSL, heterophily-oriented methods consistently outperform traditional SSL approaches on heterophilic datasets, without sacrificing performance on homophilic graphs, and they even rival or surpass supervised baselines—remarkably without using any labels.

Table 4: Computation and Memory Comparisons (Use SIMPLEX-GT (CF) for simplicity)

| Datasets | GPU MEMORY(MB) | | | | EPOCH TIME(S/EPOCH) | | | | TOTAL TIME(S) | | | |
|----------|------------|------|-------|------------|------------|------|-------|------------|------------|--------|--------|------------|
| | NODEFORMER | MUSE | GREET | SIMPLEX-GT | NODEFORMER | MUSE | GREET | SIMPLEX-GT | NODEFORMER | MUSE | GREET | SIMPLEX-GT |
| Actor | **1157** | 8785 | 4318 | 1793 | 0.08 | 0.45 | 0.58 | **0.07** | 25.20 | 53.09 | 66.71 | **14.84** |
| Roman | **4048** | 34798 | 36433 | 6863 | 0.23 | 2.48 | 2.82 | **0.20** | 103.04 | 301.35 | 379.21 | **44.62** |

## 3.2 EFFICIENCY ANALYSIS

Besides the theoretical complexity analysis in Sec.2, we present the empirical comparisons of computation and memory efficiency in Table 4. For this purpose, we include NodeFormer—a widely used full-batch sparse Transformer—as well as two strong SSL baselines, MUSE and GREET, and evaluate them on two large-scale datasets, Actor and Roman-Empire. The results with SSL baselines show that our method achieves substantial advantages over MUSE and GREET in both memory consumption and training speed. This improvement can be attributed to our novel node-to-cluster

Table 5: Results on Ablating Three Components.

| Methods | Cornell | Texas | Wisconsin | Actor | Cora | Citeseer | Pubmed | Arxiv |
|---|---|---|---|---|---|---|---|---|
| SimPlex-GT (CS) | 84.86±3.24 | 92.97±3.66 | 92.16±2.63 | 37.68±0.83 | 83.85±0.12 | 73.12±0.20 | 80.88±0.36 | 71.97±0.32 |
| w/o N2C | 82.16±2.16 | 87.83±3.67 | 88.23±2.35 | OOM | 78.80±0.15 | 69.26±0.28 | OOM | OOM |
| w/o CS | 82.70±2.48 | 88.92±4.26 | 88.56±2.77 | 36.57±0.88 | 83.80±0.10 | 72.70±0.18 | 80.00±0.33 | 70.98±0.28 |
| w/o SSL | 83.24±3.91 | 90.54±4.05 | 90.39±3.21 | 37.07±1.10 | 80.60±0.16 | 72.10±0.21 | 78.00±0.32 | 71.88±0.36 |

attention design. Compared with NodeFormer, our model incurs only a slight increase in memory usage (e.g., about 0.6 GB more on Actor), yet it runs nearly twice as fast, while also delivering **significantly better accuracy** (see Table 2).

### 3.3 ABLATION STUDY

**Components of SimPlex-GT.** In Table 5, we conduct ablations on four homophilic graphs (Cora, CiteSeer, PubMed, ArXiv) and four heterophilic graphs (Cornell, Texas, Wisconsin, Actor) by modifying one component at a time: (i) replacing N2C attention with full N2N attention, (ii) removing the CF/CS designs; (iii) replacing the SSL objective with SL on labels.

Table 6: The effects of # cluster

| # clusters | Texas | Roman | Cora |
|---|---|---|---|
| 2 | 91.89±3.46 | 79.56±0.62 | 83.80±0.18 |
| 5 | 92.97±4.39 | 80.86±0.59 | **84.05**±0.15 |
| 10 | 92.13±3.91 | 81.23±0.56 | 83.98±0.12 |
| 15 | 91.12±4.06 | **81.56**±0.68 | 83.20±0.20 |

- N2C plays the most critical role in overall performance. Without it, full node-to-node attention often runs out of memory even on medium-scale datasets. On heterophilic graphs, N2C effectively reduces structural noise (as shown in Table 1), while on homophilic graphs such as Cora and CiteSeer—where the labeling rate is low and the datasets are small—full attention tends to overfit, which can also be observed in Table 1. These limitations of N2N highlight the clear advantages of our proposed N2C design.
- Eliminating cluster smoothing (CS) also degrades performance, as the proposed learnable clustering process alone does not incorporate structural information. The cluster smoothing is essential for obtaining stable clusters that accurately capture community patterns in graphs (see Theorem 4).
- Replacing the SSL objective with purely SL also leads to performance degradation, especially on small datasets with limited labels such as Cora, CiteSeer, and PubMed. In these cases, the model tends to overfit the scarce labels and fails to learn generalizable representations.

**Number of Clusters.** Table 6 shows the ablation study on the number of clusters used in our proposed N2C attention. We can see that our proposed method is not sensitive to the number of clusters. Although the best setting for the number of clusters varies across datasets, usually a small number such as 5 is enough to achieve good performance while maintaining good efficiency.

Table 7: The effects of $r$

| Ratio $r$ | Texas | Roman | Cora |
|---|---|---|---|
| 0 | 91.89±3.89 | 80.32±0.53 | 84.02±0.11 |
| 0.3 | 90.54±3.82 | 81.23±0.63 | 83.98±0.21 |
| 0.5 | 91.89±3.89 | **81.56**±0.68 | **84.05**±0.15 |
| 0.8 | **92.97**±4.39 | 80.96±0.57 | 83.50±0.12 |
| 1 | 92.23±3.89 | 79.96±0.51 | 83.70±0.15 |

**Dynamic Masking Ratio $r$ (Eqn. 16).** We further evaluate performance under different dynamic masking ratios across datasets (Table 7). While the optimal ratio varies by dataset, dynamic masking consistently outperforms pure random masking ($r = 0$), highlighting the necessity of our design.

**Orthogonality Regularization** An additional study on $\lambda_{\mathrm{orth}}$ is provided in Appendix F.

## 4 CONCLUSION

In this work, we presented SimPlex-GT, a novel and efficient Graph Transformer architecture that addresses the challenges of learning on complex graphs exhibiting both homophily and heterophily. By integrating a global node-to-cluster attention mechanism with a local GCN-based message passing branch, SimPlex-GT effectively captures both coarse semantic patterns and fine-grained structural dependencies. Two complementary fusion strategies—Complementary Filtering (CF) and Cluster Smoothing (CS)—offer flexible ways to encode local-global synergies, while the orthogonality regularization promote representational diversity. Unlike prior approaches that are often tailored to either homophilic or heterophilic graphs, SimPlex-GT operates as a unified, modular framework that adapts seamlessly to diverse structural regimes. Importantly, it achieves linear complexity via N2C attention, making it scalable to large graphs without sacrificing accuracy. Empirical results on 11 benchmark datasets demonstrate that SimPlex-GT achieves state-of-the-art performance on heterophilic graphs, while remaining highly competitive on homophilic graphs. These gains are achieved with superior training efficiency and memory usage compared to existing Graph Transformers and GNNs.

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

# A RELATED WORK

## A.1 GRAPH TRANSFORMERS

The integration of Transformer architectures with graph-structured data has emerged as a promising research direction, aiming to leverage the powerful representation learning capabilities of attention mechanisms for graph neural networks. Graph Transformers (GTs) leverage pairwise attention across all node pairs, rather than restricting message passing to directly connected nodes as in traditional GNNs. In other words, global attention can be viewed as a generalization of GNN message passing to a fully connected graph, enabling the model to naturally capture long-range dependencies and potential unconnected links.

However, full node-to-node attention incurs quadratic complexity in the number of nodes ($\mathcal{O}(N^2)$), which quickly becomes prohibitive on medium–large graphs. To improve scalability, recent work introduces *sparse* GTs that approximate or restructure attention to sub-quadratic complexity while retaining global receptive fields. For example, GraphGPS (Rampášek et al., 2022) proposes a hybrid design that combines local message passing with a global sparse attention block and structural encodings, which balances inductive bias but introduces complexity overhead and dependence on positional encodings. NodeFormer (Wu et al., 2022a) employs random feature approximations to compute kernelized attention in nearly linear time, offering scalability but at the cost of approximation error and sensitivity to hyperparameters. Exphormer (Shirzad et al., 2023) leverages expander graphs to enforce structured sparsity with constant-degree connections, achieving global reachability at near-linear cost. GOAT (Kong et al., 2023) introduces adaptive tokenization to compress graphs into a smaller set of latent tokens, reducing computation and enabling semantic abstraction, but potentially losing information if tokenization is suboptimal. Similarly, SGFormer (Wu et al., 2023) exploits spectral sparsification to approximate global interactions more efficiently, grounded in spectral theory but sensitive to eigen-structure and costly on very large graphs. Finally, NAGphormer (Chen et al., 2022b) constrains attention to neighborhood-aware patterns, which improves stability on small datasets but may limit expressiveness by missing long-range dependencies.

Overall, sparse GTs vary in their strategies—ranging from kernel approximation and hybridization to structured sparsity, tokenization, or spectral priors—but they share the strengths of scalability and global context while facing challenges such as approximation bias, hyperparameter sensitivity, and reliance on structural encodings, which can lead to suboptimal performance.

## A.2 LEARNING ON HETEROPHILIC GRAPHS

Heterophilic graphs commonly arise in diverse real-world scenarios, including online fraud detection networks (Pandit et al., 2007), dating platforms (Altenburger & Ugander, 2018), and molecular interaction graphs (Zhu et al., 2020a). In such graphs, linked nodes often have dissimilar attributes and may belong to different classes, posing challenges for conventional GNNs. This has led to a growing body of research on developing architectures that can better capture and propagate information in heterophilic settings.

One line of work explores aggregating signals from long-range or higher-order neighbors (Li et al., 2022; Liu et al., 2021; Abu-El-Haija et al., 2019; Pei et al., 2020; Suresh et al., 2021). For example, MixHop (Abu-El-Haija et al., 2019) collects and concatenates information from multiple hop distances at each layer, while Geom-GCN (Pei et al., 2020) establishes neighborhood relationships in a continuous latent space rather than the raw topology. Similarly, WRGAT (Suresh et al., 2021) reconstructs computation graphs by reweighting and redefining edges across the entire network, enabling information flow from more distant nodes.

Another strand focuses on rethinking the design of message passing in GNNs (Chen et al., 2020; Chien et al., 2020; Yan et al., 2021; Zhu et al., 2020a). GPR-GNN (Chien et al., 2020), for instance, leverages the Generalized PageRank formulation to assign learnable propagation weights across layers, whereas H2GCN (Zhu et al., 2020a) explicitly removes self-loops and introduces non-mixing mechanisms to better preserve ego-node representations.

A complementary direction builds on spectral graph analysis (Luan et al., 2021; Bo et al., 2021), emphasizing that high-pass filtering can be advantageous under heterophily by amplifying differences across neighbors and preserving high-frequency components in node signals.

Despite these advances, most approaches still rely strongly on abundant labeled data, which is rarely feasible in practice due to the cost of annotation and the difficulty of maintaining label quality. Moreover, their reliance on supervision restricts their ability to autonomously extract structural and feature patterns from the graph.

### A.3 Graph Representation Learning via SSL

Self-supervised learning (SSL) has recently become a dominant paradigm for graph representation learning, generally categorized into *contrastive* and *generative* approaches.

**Graph contrastive learning** Contrastive methods learn node or graph representations by maximizing the agreement between multiple graph views. Early approaches such as DGI (Veličković et al., 2018) and MVGRL (Hassani & Khasahmadi, 2020) capture both local and global signals, while GRACE (Zhu et al., 2020b) applies augmentation-based contrast at the node level. Among recent advances on heterophilic graphs, HGRL (Chen et al., 2022a) reconstructs similarity matrices for dual feature augmentations, S3GCL (Wan et al., 2024) employs spectral–spatial contrastive learning to efficiently capture multi-scale graph signals, but it still relies on negative sampling and careful design of spectral filters. GREET (Liu et al., 2022) separates homophilic and heterophilic edges for low/high-pass filtering, and MUSE (Yuan et al., 2023) perturbs both topology and features with a structure-based combiner. These heterophily-aware SSL methods consistently outperform traditional contrastive approaches, though many still depend on careful negative sampling or alternating training schemes, leading to high computational overhead and potential suboptimal performance.

**Graph generative learning** Generative methods instead reconstruct graph features or structures. Classical approaches (e.g., GAE, VGAE, MGAE) mainly focus on topology reconstruction, while GraphMAE (Hou et al., 2022) adopt masked feature modeling and have set strong baselines on homophilic graphs. For heterophilic settings, DSSL (Xiao et al., 2022) decouples diverse structural patterns. Despite these innovations, generative methods on heterophilic graphs still lag behind state-of-the-art contrastive SSL methods (Hou et al., 2022).

## B Proof of Theorem 1

### B.1 Definition and Setup

Consider a node classification task on graph $\mathcal{G} = (\mathcal{V}, \mathcal{E})$ with node features $\mathbf{X} \in \mathbb{R}^{n \times d}$ and labels $\mathbf{y} \in \{-1, +1\}^n$.

**Definition 1** (Node-level Homophily). *For node $i \in \mathcal{V}$, define its local homophily ratio as:*

$$\rho_i = \frac{|\{j \in \mathcal{N}(i) : y_j = y_i\}|}{|\mathcal{N}(i)|} \tag{17}$$

*Node $i$ is locally heterophilic if $\rho_i < 0.5$.*

**Definition 2** (Heterophilic Node Set). *Define the set of heterophilic nodes as:*

$$\mathcal{V}_{hetero} = \{i \in \mathcal{V} : \rho_i < 0.5\} \tag{18}$$

### B.2 Main Theorem

**Theorem 1.** *Suppose a graph with a significant portion of nodes are locally heterophilic, under the assumption that initial features are perfectly aligned with labels (i.e., $\mathbf{x}_i^{(0)} = y_i \mathbf{e}$ for some unit vector $\mathbf{e}$), we have:*

*1. **GNN:** For any node $i \in \mathcal{V}_{hetero}$:*

$$\langle \mathbf{h}_i^{(1)}, y_i \mathbf{e} \rangle = (2\rho_i - 1)||\mathbf{e}||^2 < 0 \tag{19}$$

*2. **Graph Transformer:** There exists parameters $\boldsymbol{\theta}^*$ such that:*

$$\langle \mathbf{h}_i^{(1)}, y_i \mathbf{e} \rangle > 0, \quad \forall i \in \mathcal{V} \tag{20}$$

*Proof.* **GNN:**

For a standard GNN with mean aggregation at node $i$:

$$\mathbf{h}_i^{(1)} = \sigma\left(\frac{1}{|\mathcal{N}(i)|} \sum_{j \in \mathcal{N}(i)} \mathbf{W}^{(0)} \mathbf{x}_j^{(0)}\right) \tag{21}$$

With linear activation and $\mathbf{W}^{(0)} = \mathbf{I}$:

$$\mathbf{h}_i^{(1)} = \frac{1}{|\mathcal{N}(i)|} \sum_{j \in \mathcal{N}(i)} y_j \mathbf{e} \tag{22}$$

Decomposing the neighborhood into same-class and different-class neighbors:

$$\mathbf{h}_i^{(1)} = \frac{1}{|\mathcal{N}(i)|} \left(\sum_{j \in \mathcal{N}(i): y_j = y_i} y_i \mathbf{e} + \sum_{j \in \mathcal{N}(i): y_j \neq y_i} (-y_i)\mathbf{e}\right) \tag{23}$$

$$= \frac{1}{|\mathcal{N}(i)|} \left(\rho_i |\mathcal{N}(i)| \cdot y_i \mathbf{e} + (1 - \rho_i)|\mathcal{N}(i)| \cdot (-y_i)\mathbf{e}\right) \tag{24}$$

$$= y_i(2\rho_i - 1)\mathbf{e} \tag{25}$$

Therefore:

$$\langle \mathbf{h}_i^{(1)}, y_i \mathbf{e}\rangle = y_i^2(2\rho_i - 1)\|\mathbf{e}\|^2 = (2\rho_i - 1)\|\mathbf{e}\|^2 \tag{26}$$

For heterophilic nodes where $\rho_i < 0.5$: $(2\rho_i - 1) < 0$, thus $\langle \mathbf{h}_i^{(1)}, y_i \mathbf{e}\rangle < 0$.

**Graph Transformer:** Set

$$\mathbf{W}_Q = \mathbf{W}_K = \gamma\,\mathbf{I}, \qquad \mathbf{W}_V = \mathbf{I}, \quad \gamma > 0. \tag{27}$$

Then

$$\alpha_{ij} \propto \exp\left(\frac{(\mathbf{W}_Q \mathbf{x}_i^{(0)})^T (\mathbf{W}_K \mathbf{x}_j^{(0)})}{\sqrt{d}}\right) = \exp\left(\frac{\gamma^2 (\mathbf{x}_i^{(0)})^T \mathbf{x}_j^{(0)}}{\sqrt{d}}\right) = \exp\left(\frac{\gamma^2 y_i y_j \|\mathbf{e}\|^2}{\sqrt{d}}\right). \tag{28}$$

Hence

$$\alpha_{ij} \propto \begin{cases} \exp\left(\dfrac{\gamma^2\|\mathbf{e}\|^2}{\sqrt{d}}\right) & \text{if } y_i = y_j, \\ \exp\left(-\dfrac{\gamma^2\|\mathbf{e}\|^2}{\sqrt{d}}\right) & \text{if } y_i \neq y_j. \end{cases} \tag{29}$$

**Sharpening limit via $\gamma$.** Split the normalizer into same/different-label parts and divide through by $\exp(\gamma^2\|\mathbf{e}\|^2/\sqrt{d})$:

$$\alpha_{ij} = \begin{cases} \dfrac{1}{|\{k : y_k = y_i\}| + |\{k : y_k \neq y_i\}|\,\exp\left(-\frac{2\gamma^2\|\mathbf{e}\|^2}{\sqrt{d}}\right)} & \text{if } y_j = y_i, \\ beginequation12pt]\dfrac{\exp\left(-\frac{2\gamma^2\|\mathbf{e}\|^2}{\sqrt{d}}\right)}{|\{k : y_k = y_i\}| + |\{k : y_k \neq y_i\}|\,\exp\left(-\frac{2\gamma^2\|\mathbf{e}\|^2}{\sqrt{d}}\right)} & \text{if } y_j \neq y_i. \end{cases} \tag{30}$$

As $\gamma \to \infty$ (effective sharpening increases), we have $\exp\left(-\frac{2\gamma^2\|\mathbf{e}\|^2}{\sqrt{d}}\right) \to 0$, and therefore

$$\alpha_{ij} \to \begin{cases} \dfrac{1}{|\{k : y_k = y_i\}|} & \text{if } y_i = y_j, \\ 0 & \text{if } y_i \neq y_j. \end{cases} \tag{31}$$

Consequently,

$$\mathbf{h}_i^{(1)} = \sum_{j \in \mathcal{V}} \alpha_{ij}\,\mathbf{x}_j^{(0)} = \sum_{j: y_j = y_i} \alpha_{ij}\,y_j \mathbf{e} = y_i \mathbf{e} \sum_{j: y_j = y_i} \alpha_{ij} = y_i \mathbf{e}, \tag{32}$$

and thus

$$\langle \mathbf{h}_i^{(1)}, y_i \mathbf{e}\rangle = \|\mathbf{e}\|^2 > 0. \tag{33}$$

$$\square$$

## B.3 PERFORMANCE BOUNDS

**Corollary 1.** *Define the heterophilic node set $\mathcal{V}_{hetero} = \{ i : \rho_i < \frac{1}{2} \}$. Under the conditions of Theorem 1:*

1. ***GNN:** A one-layer mean-aggregation (linearized) GNN misclassifies every node with $\rho_i < \frac{1}{2}$; hence it misclassifies at least a $|\mathcal{V}_{hetero}|/|\mathcal{V}|$ fraction of nodes.*

2. ***Graph Transformer:** With the attention in Theorem 1 (i.e., the sharpening limit), the first-layer outputs satisfy $\langle \mathbf{h}_i^{(1)}, y_i \mathbf{e} \rangle > 0$ for all $i$, yielding zero classification error.*

## C PROOF OF THEOREM 2

**Theorem 2** (Variance reduction via soft clusters). *Assume $x_i^{(0)} = y_i \mathbf{e} + \varepsilon_i$ with $\mathbb{E}[\varepsilon_i] = 0$ and $\text{Cov}(\varepsilon_i) = \Sigma \preceq \sigma^2 I$. Let the soft assignment matrix be $p_{i,c} \in (0,1)$ with $\sum_{c=1}^{C} p_{i,c} = 1$ for each node $i$. Define the (soft) cluster mass and prototype by*

$$s_c \triangleq \sum_{i=1}^{N} p_{i,c}, \tag{34}$$

$$\mathbf{p}_c \triangleq \frac{1}{s_c} \sum_{i=1}^{N} p_{i,c} \, x_i^{(0)}. \tag{35}$$

*Let the Node-to-Cluster (N2C) output be $h_i^{\text{N2C}} \triangleq \sum_{c=1}^{C} \alpha_{i \to c} \, \mathbf{p}_c$ where $\alpha_{i \to c} \geq 0$ and $\sum_{c=1}^{C} \alpha_{i \to c} = 1$. Then, along direction $\mathbf{e}$, the following bounds hold (without assuming independence across clusters):*

$$\text{Var}(\langle \mathbf{p}_c, \mathbf{e} \rangle) \leq \sigma^2 \|\mathbf{e}\|^2 \cdot \frac{\sum_{i=1}^{N} p_{i,c}^2}{\left(\sum_{i=1}^{N} p_{i,c}\right)^2}, \tag{36}$$

$$\text{Var}(\langle h_i^{\text{N2C}}, \mathbf{e} \rangle) \leq \sum_{c=1}^{C} \alpha_{i \to c} \, \sigma^2 \|\mathbf{e}\|^2 \cdot \frac{\sum_{i=1}^{N} p_{i,c}^2}{\left(\sum_{i=1}^{N} p_{i,c}\right)^2}. \tag{37}$$

*By contrast, a single-neighbor Node-to-Node readout satisfies*

$$\text{Var}(\langle x_j^{(0)}, \mathbf{e} \rangle) = \text{Var}(\langle \varepsilon_j, \mathbf{e} \rangle) = \mathbf{e}^\top \Sigma \mathbf{e} \leq \sigma^2 \|\mathbf{e}\|^2. \tag{38}$$

*Proof.* **Step 1: Reweighting form of the prototype.** Let $w_{i,c} \triangleq p_{i,c}/s_c$ so that $\sum_{i=1}^{N} w_{i,c} = 1$. Then

$$\mathbf{p}_c = \sum_{i=1}^{N} w_{i,c} \, x_i^{(0)} = \sum_{i=1}^{N} w_{i,c} \, (y_i \mathbf{e} + \varepsilon_i), \tag{39}$$

$$\langle \mathbf{p}_c, \mathbf{e} \rangle = \sum_{i=1}^{N} w_{i,c} \, (y_i \|\mathbf{e}\|^2 + \langle \varepsilon_i, \mathbf{e} \rangle). \tag{40}$$

The class-mean term $\sum_i w_{i,c} y_i \|\mathbf{e}\|^2$ is deterministic w.r.t. noise, hence

$$\text{Var}(\langle \mathbf{p}_c, \mathbf{e} \rangle) = \text{Var}\left( \sum_{i=1}^{N} w_{i,c} \, \langle \varepsilon_i, \mathbf{e} \rangle \right). \tag{41}$$

**Step 2: Bounding the prototype variance.** Assuming independence across nodes and $\text{Cov}(\varepsilon_i) = \Sigma \preceq \sigma^2 I$,

$$\text{Var}\left( \sum_{i=1}^{N} w_{i,c} \, \langle \varepsilon_i, \mathbf{e} \rangle \right) = \sum_{i=1}^{N} w_{i,c}^2 \, \text{Var}(\langle \varepsilon_i, \mathbf{e} \rangle) = (\mathbf{e}^\top \Sigma \mathbf{e}) \sum_{i=1}^{N} w_{i,c}^2, \tag{42}$$

$$\text{Var}(\langle \mathbf{p}_c, \mathbf{e} \rangle) \leq \sigma^2 \|\mathbf{e}\|^2 \sum_{i=1}^{N} w_{i,c}^2 = \sigma^2 \|\mathbf{e}\|^2 \cdot \frac{\sum_{i=1}^{N} p_{i,c}^2}{\left(\sum_{i=1}^{N} p_{i,c}\right)^2}. \tag{43}$$

**Step 3: N2C output as a convex combination (correlated clusters).** We have

$$\langle h_i^{\text{N2C}}, \mathbf{e}\rangle = \sum_{c=1}^{C} \alpha_{i\to c}\, \langle \mathbf{p}_c, \mathbf{e}\rangle, \qquad \alpha_{i\to c} \ge 0,\ \sum_{c=1}^{C} \alpha_{i\to c} = 1. \tag{44}$$

Without assuming independence across $\langle \mathbf{p}_c, \mathbf{e}\rangle$, expand and bound via Cauchy–Schwarz:

$$\text{Var}\Big( \sum_{c=1}^{C} \alpha_{i\to c}\, \langle \mathbf{p}_c, \mathbf{e}\rangle \Big) \le \Big( \sum_{c=1}^{C} \alpha_{i\to c}\, \sqrt{\text{Var}(\langle \mathbf{p}_c, \mathbf{e}\rangle)} \Big)^2. \tag{45}$$

From Step 2 we already know

$$\text{Var}(\langle \mathbf{p}_c, \mathbf{e}\rangle) \le \sigma^2 \|\mathbf{e}\|^2 \cdot \frac{\sum_i p_{i,c}^2}{\left( \sum_i p_{i,c} \right)^2}. \tag{46}$$

Substituting into the previous bound gives

$$\text{Var}\big( \langle h_i^{\text{N2C}}, \mathbf{e}\rangle \big) \le \left( \sum_{c=1}^{C} \alpha_{i\to c}\, \sigma \|\mathbf{e}\| \cdot \frac{\sqrt{\sum_i p_{i,c}^2}}{\sum_i p_{i,c}} \right)^2. \tag{47}$$

Using $\sqrt{\text{Var}(\langle \mathbf{p}_c, \mathbf{e}\rangle)} \le \sigma \|\mathbf{e}\| \cdot \sqrt{\sum_{i=1}^{N} p_{i,c}^2} / \left( \sum_{i=1}^{N} p_{i,c} \right)$ and Jensen/Cauchy–Schwarz with $\sum_c \alpha_{i\to c} = 1$, we obtain

$$\text{Var}\big( \langle h_i^{\text{N2C}}, \mathbf{e}\rangle \big) \le \sum_{c=1}^{C} \alpha_{i\to c}\, \sigma^2 \|\mathbf{e}\|^2 \cdot \frac{\sum_{i=1}^{N} p_{i,c}^2}{\left( \sum_{i=1}^{N} p_{i,c} \right)^2}. \tag{48}$$

**Step 4: N2N (single neighbor) comparison.** For any single-neighbor readout,

$$\text{Var}\big( \langle x_j^{(0)}, \mathbf{e}\rangle \big) = \text{Var}\big( \langle \varepsilon_j, \mathbf{e}\rangle \big) = \mathbf{e}^\top \Sigma \mathbf{e} \le \sigma^2 \|\mathbf{e}\|^2. \tag{49}$$

Let

$$R \triangleq \sum_{c=1}^{C} \alpha_{i\to c}\, \sigma^2 \|\mathbf{e}\|^2\, \frac{\sum_{j=1}^{N} p_{j,c}^2}{\left( \sum_{j=1}^{N} p_{j,c} \right)^2} = \sigma^2 \|\mathbf{e}\|^2 \sum_{c=1}^{C} \alpha_{i\to c}\, r_c, \qquad r_c \triangleq \frac{\sum_j p_{j,c}^2}{s_c^2},\ s_c \triangleq \sum_j p_{j,c}. \tag{50}$$

Then $R$ is the variance upper bound from Step 3. We now show why $R \le \sigma^2 \|\mathbf{e}\|^2$.

**Bounds on $r_c$.** Define normalized weights $w_{j,c} = \frac{p_{j,c}}{s_c}$. Then:

- $w_{j,c} \ge 0$ and $\sum_j w_{j,c} = 1$ (forms a probability distribution)

- $0 \le w_{j,c} \le 1$ for all $j$ (since $p_{j,c} \le s_c$ always holds)

Therefore:

$$r_c = \frac{\sum_j p_{j,c}^2}{s_c^2} = \sum_j \left( \frac{p_{j,c}}{s_c} \right)^2 = \sum_j w_{j,c}^2 \tag{51}$$

Since $0 \le w_{j,c} \le 1$, we have $w_{j,c}^2 \le w_{j,c}$, thus:

$$r_c = \sum_j w_{j,c}^2 \le \sum_j w_{j,c} = 1 \tag{52}$$

**Convex combination.** Since $\alpha_{i\to c} \ge 0$ and $\sum_c \alpha_{i\to c} = 1$ (attention weights form a probability distribution):

$$\sum_{c=1}^{C} \alpha_{i\to c} \cdot r_c \le \sum_{c=1}^{C} \alpha_{i\to c} \cdot 1 = 1 \tag{53}$$

Hence:

$$R = \sigma^2 \|\mathbf{e}\|^2 \sum_c \alpha_{i\to c} r_c \le \sigma^2 \|\mathbf{e}\|^2 \tag{54}$$

The inequality $r_c < 1$ is strict whenever at least two nodes contribute to cluster $c$ □

**Remark 2.** *The variance reduction factor $r_c \approx 1/n_{eff}$ where $n_{eff}$ is the effective number of nodes in cluster $c$. Thus N2C provides denoising proportional to cluster sizes, while N2N lacks such benefit.*

*Proof.* Let $s_c := \sum_i p_{i,c}$ and $w_{i,c} := p_{i,c}/s_c$ so that $\sum_i w_{i,c} = 1$. Then

$$r_c = \frac{\sum_i p_{i,c}^2}{\left(\sum_i p_{i,c}\right)^2} = \sum_i w_{i,c}^2 = 1/n_{\text{eff}} \tag{55}$$

Hence:

$$\text{Var}(\langle \mathbf{p}_c, \mathbf{e} \rangle) \ \leq \ (\mathbf{e}^\top \Sigma \mathbf{e}) \, r_c \ \leq \ \sigma^2 \|\mathbf{e}\|^2 \frac{1}{n_{\text{eff}}}. \tag{56}$$

Consequently,

$$\text{Var}\Big(\sum_c \alpha_{i\to c} \langle \mathbf{p}_c, \mathbf{e} \rangle\Big) \ \leq \ \sigma^2 \|\mathbf{e}\|^2 \sum_c \alpha_{i\to c} \, r_c \ = \ \sigma^2 \|\mathbf{e}\|^2 \sum_c \alpha_{i\to c} \frac{1}{n_{\text{eff},c}}. \tag{57}$$

Thus N2C provides denoising proportional to cluster effective sizes ($n_{\text{eff}}$), whereas N2N has no such $1/n_{\text{eff}}$ factor. □

## D  PROOF OF THEOREM 3

**Lemma 1** (Near-orthogonality in $\ell_2$)**.** *Let $L = U\Lambda U^\top$ with $U^\top U = I$, and define $G = U\,g(\Lambda)\,U^\top$, $H = I - G = U\,(I - g(\Lambda))\,U^\top$ for a response $g : [0,\infty) \to [0,1]$. For $x \in \mathbb{R}^{N\times d}$, write $\hat{X} := U^\top x$ (its $k$-th row is $\hat{x}_k^\top$). Then*

$$\langle x_g, x_f \rangle := \text{Tr}(x_g^\top x_f) = \sum_{k=1}^N g(\lambda_k)\big(1 - g(\lambda_k)\big)\|\hat{x}_k\|_2^2 \ \leq \ \varepsilon\,\|x\|_F^2, \tag{58}$$

*where $\varepsilon := \max_{\lambda \geq 0} g(\lambda)(1 - g(\lambda)) \leq \frac{1}{4}$, and equality $\langle x_g, x_f \rangle = 0$ holds if $g(\lambda) \in \{0,1\}$ for all $\lambda$ (spectral projector).*

*Proof.* By definition $x_g = Gx = U\,g(\Lambda)\,\hat{X}$ and $x_f = Hx = U\,(I - g(\Lambda))\,\hat{X}$. Using $U^\top U = I$ and cyclicity of trace,

$$\langle x_g, x_f \rangle = \text{Tr}\Big((Ug(\Lambda)\hat{X})^\top (U(I - g(\Lambda))\hat{X})\Big) = \text{Tr}\Big(\hat{X}^\top g(\Lambda)(I - g(\Lambda))\hat{X}\Big)$$
$$= \sum_{k=1}^N g(\lambda_k)\big(1 - g(\lambda_k)\big)\|\hat{x}_k\|_2^2, \tag{59}$$

since $g(\Lambda)$ and $I - g(\Lambda)$ are diagonal and multiply elementwise in the eigenbasis. Because $0 \leq g(\lambda) \leq 1$, we have $0 \leq g(\lambda)(1 - g(\lambda)) \leq \varepsilon \leq \frac{1}{4}$, hence

$$|\langle x_g, x_f \rangle| \leq \varepsilon \sum_{k=1}^N \|\hat{x}_k\|_2^2 = \varepsilon\,\|x\|_F^2, \tag{60}$$

using $\|x\|_F^2 = \|\hat{X}\|_F^2$ (orthonormal $U$ preserves the Frobenius norm). If $g$ is a spectral projector, each factor $g(\lambda_k)(1 - g(\lambda_k)) = 0$, so $\langle x_g, x_f \rangle = 0$. □

**Lemma 2** (Variance reduction via contraction)**.** *Let $\eta \in \mathbb{R}^N$ be a zero-mean random vector with covariance $\text{Cov}(\eta) \preceq \sigma^2 I$. For any unit $\mathbf{e} \in \mathbb{R}^N$,*

$$\text{Var}\big(\langle G\eta, \mathbf{e} \rangle\big) \ = \ \mathbf{e}^\top G\,\text{Cov}(\eta)\,G\,\mathbf{e} \ \leq \ \sigma^2\,\mathbf{e}^\top G^2 \mathbf{e} \ \leq \ \sigma^2\,\mathbf{e}^\top G \mathbf{e} \ \leq \ \sigma^2. \tag{61}$$

*The inequalities are strict whenever $g(\lambda) \in (0,1)$ on a set of eigenvalues with nonzero variance mass along $v$. An analogous statement holds for $H = I - G$.*

*Proof.* First, $\langle G\eta, \mathbf{e} \rangle = \langle \eta, G^\top \mathbf{e} \rangle$ and $G$ is symmetric, so

$$\text{Var}(\langle G\eta, \mathbf{e} \rangle) = \text{Var}(\langle \eta, G\mathbf{e} \rangle) = (G\mathbf{e})^\top \text{Cov}(\eta)\,(G\mathbf{e}) = \mathbf{e}^\top G\,\text{Cov}(\eta)\,G\,\mathbf{e}. \tag{62}$$

Using $\text{Cov}(\eta) \preceq \sigma^2 I$ gives $v^\top G\,\text{Cov}(\eta)\,G\,v \leq \sigma^2\,v^\top G^2 v$.

Now write $G = U\, g(\Lambda)\, U^\top$. Then

$$\mathbf{e}^\top G^2 \mathbf{e} = \mathbf{e}^\top U g(\Lambda)^2 U^\top \mathbf{e} = \sum_{k=1}^N g(\lambda_k)^2\, \xi_k^2, \qquad \mathbf{e}^\top G \mathbf{e} = \sum_{k=1}^N g(\lambda_k)\, \xi_k^2, \qquad (63)$$

where $\xi = U^\top \mathbf{e}$ and $\sum_k \xi_k^2 = \|\mathbf{e}\|^2 = 1$. Since $0 \le g(\lambda) \le 1$, we have $g(\lambda)^2 \le g(\lambda)$, hence $\mathbf{e}^\top G^2 \mathbf{e} \le \mathbf{e}^\top G \mathbf{e} \le \sum_k \xi_k^2 = 1$. Combining the steps yields the chain of inequalities.

Strictness: if there exists $k$ with $\xi_k \ne 0$, $g(\lambda_k) \in (0,1)$ and positive variance along that eigen-direction (i.e., the bound $\mathrm{Cov}(\eta) \preceq \sigma^2 I$ is not tight as zero there), then at least one inequality becomes strict. The same argument applies to $H = U(I - g(\Lambda))U^\top$ since its eigenvalues lie in $[0,1]$ and are $1 - g(\lambda)$. $\qquad\square$

**Theorem 3** (Stability of N2C attention logits). *Let keys/values be computed from low-pass prototypes and queries from high-pass features:*

$$s_{i,c} := \frac{1}{\sqrt{d_k}} \left\langle Q(x_f^{(i)}),\, K(c_b^{(c)}) \right\rangle, \qquad s_{i,c}^\star := \frac{1}{\sqrt{d_k}} \left\langle Q(h^{(i)}),\, K(s^{(c)}) \right\rangle, \qquad (64)$$

*where $h^{(i)}$ is the high-frequency component of node $i$, and $s^{(c)}$ the low-frequency prototype of cluster $c$. Assume $Q$ and $K$ are Lipschitz: $\|Q(a) - Q(b)\| \le L_Q \|a - b\|$, $\|K(a) - K(b)\| \le L_K \|a - b\|$, and that $\mathbb{E}\|K(c_b^{(c)})\|^2 \le M_K^2$, $\mathbb{E}\|Q(h^{(i)})\|^2 \le M_Q^2$. Then*

$$\mathbb{E}\left[ (s_{i,c} - s_{i,c}^\star)^2 \right] \le \frac{2}{d_k} \left( L_Q^2 M_K^2\, \mathbb{E}\|x_f^{(i)} - h^{(i)}\|^2 + L_K^2 M_Q^2\, \mathbb{E}\|c_b^{(c)} - s^{(c)}\|^2 \right). \qquad (65)$$

*Consequently, by Lemma 2, both error terms on the right shrink under the contractions $H$ and $G$, yielding strictly smaller logit MSE than using unsplit features whenever the filters attenuate off-target spectra.*

*Proof.* Let $a := Q(x_f^{(i)})$, $a^\star := Q(h^{(i)})$, $b := K(c_b^{(c)})$, $b^\star := K(s^{(c)})$. Then

$$s_{i,c} - s_{i,c}^\star = \frac{1}{\sqrt{d_k}} \left( \langle a, b \rangle - \langle a^\star, b^\star \rangle \right) \qquad (66)$$

$$= \frac{1}{\sqrt{d_k}} \Big( \underbrace{\langle a, b \rangle - \langle a^\star, b \rangle}_{= \langle a - a^\star,\, b \rangle} + \underbrace{\langle a^\star, b \rangle - \langle a^\star, b^\star \rangle}_{= \langle a^\star,\, b - b^\star \rangle} \Big) \qquad (67)$$

$$= \frac{1}{\sqrt{d_k}} \Big( \langle a - a^\star,\, b \rangle + \langle a^\star,\, b - b^\star \rangle \Big). \qquad (68)$$

By Cauchy–Schwarz,

$$|s_{i,c} - s_{i,c}^\star| \le \frac{1}{\sqrt{d_k}} \big( \|a - a^\star\|\, \|b\| + \|a^\star\|\, \|b - b^\star\| \big). \qquad (69)$$

Square and use $(u+v)^2 \le 2(u^2 + v^2)$:

$$(s_{i,c} - s_{i,c}^\star)^2 \le \frac{2}{d_k} \big( \|a - a^\star\|^2\, \|b\|^2 + \|a^\star\|^2\, \|b - b^\star\|^2 \big). \qquad (70)$$

Apply Lipschitz bounds $\|a - a^\star\| \le L_Q \|x_f^{(i)} - h^{(i)}\|$ and $\|b - b^\star\| \le L_K \|c_b^{(c)} - s^{(c)}\|$, then take expectation and use $\mathbb{E}\|b\|^2 \le M_K^2$, $\mathbb{E}\|a^\star\|^2 \le M_Q^2$:

$$\mathbb{E}(s_{i,c} - s_{i,c}^\star)^2 \le \frac{2}{d_k} \Big( L_Q^2\, \mathbb{E}\|x_f^{(i)} - h^{(i)}\|^2\, \mathbb{E}\|b\|^2 + L_K^2\, \mathbb{E}\|c_b^{(c)} - s^{(c)}\|^2\, \mathbb{E}\|a^\star\|^2 \Big), \qquad (71)$$

which yields the stated bound. Finally, Lemma 2 (applied to $H$ on $x$ and to $G$ on prototypes) implies that both $\mathbb{E}\|x_f^{(i)} - h^{(i)}\|^2$ and $\mathbb{E}\|c_b^{(c)} - s^{(c)}\|^2$ strictly decrease whenever the corresponding filters have eigenvalues strictly inside $(0,1)$ on nontrivial spectral mass, so the right-hand side is strictly smaller than without complementary filtering. $\qquad\square$

## E    PROOF OF THEOREM 4

**Theorem 4** (Variance reduction). *Let $C$ be the number of soft clusters. For each cluster $c$, where $c \notin \mathcal{N}(c)$, unsmoothed prototype $z_c \in \mathbb{R}^d$ is written as $z_c = \mu_c + \varepsilon_c$, where $\mu_c$ is the cluster mean and $\varepsilon_c$ is a zero-mean random vector with $\mathbb{E}[\varepsilon_c] = 0$ and $\mathrm{Cov}(\varepsilon_c) \preceq \sigma^2 I$. Let $\mathcal{N}(c)$ denote neighbors of $c$ on the cluster graph $G_c$ (e.g., $A_c = P^\top A P$). Let weights $w_{cu} \ge 0$ satisfy $\sum_{u \in \mathcal{N}(c)} w_{cu} = 1$.*

*We consider one-step residual low-pass smoothing: $\tilde{z}_c = (1-\alpha)z_c + \alpha \sum_{u \in \mathcal{N}(c)} w_{cu} z_u$, where $\alpha \in (0,1)$. and $v \in \mathbb{R}^d$ denotes any unit vector, i.e., $\|\mathbf{e}\| = 1$. Assume that $\{\varepsilon_u\}_{u=1}^C$ are independent across clusters. Then, for any unit direction $\mathbf{e}$,*

$$\mathrm{Var}\big(\langle \tilde{z}_c - \mu_c, \mathbf{e}\rangle\big) = \mathrm{Var}\Big(\big\langle (1-\alpha)\varepsilon_c + \alpha \sum_u w_{cu}\varepsilon_u, \mathbf{e}\big\rangle\Big) \leq \big((1-\alpha)^2 + \alpha^2\big)\sigma^2 \leq \sigma^2, \quad (72)$$

*with strict inequality whenever $0 < \alpha < 1$.*

*Proof.* By the residual low-pass update and the noise model $z_c = \mu_c + \varepsilon_c$, $z_u = \mu_u + \varepsilon_u$,

$$\tilde{z}_c - \mu_c = (1-\alpha)(\mu_c + \varepsilon_c) + \alpha \sum_u w_{cu}(\mu_u + \varepsilon_u) - \mu_c$$

$$= \underbrace{\alpha \sum_u w_{cu}(\mu_u - \mu_c)}_{=: \, b_c \, \text{(structure bias)}} + \underbrace{(1-\alpha)\varepsilon_c + \alpha \sum_u w_{cu}\varepsilon_u}_{\text{noise part}}. \quad (73)$$

Projecting onto an arbitrary unit direction $\mathbf{e} \in \mathbb{R}^d$ turns vectors into scalars. Define

$$X := \langle \varepsilon_c, \mathbf{e}\rangle, \qquad Y := \big\langle \sum_u w_{cu}\varepsilon_u, \, \mathbf{e}\big\rangle, \qquad \beta := \langle b_c, \mathbf{e}\rangle = \alpha \sum_u w_{cu}\langle \mu_u - \mu_c, \mathbf{e}\rangle. \quad (74)$$

Then we have the exact decomposition

$$\langle \tilde{z}_c - \mu_c, \mathbf{e}\rangle = (1-\alpha)X + \alpha Y + \beta. \quad (75)$$

Since $\beta$ is a constant (given the means), it does not affect variance:

$$\mathrm{Var}\big(\langle \tilde{z}_c - \mu_c, \mathbf{e}\rangle\big) = \mathrm{Var}\big((1-\alpha)X + \alpha Y\big). \quad (76)$$

Using the variance decomposition formula gives

$$\mathrm{Var}\big((1-\alpha)X + \alpha Y\big) = (1-\alpha)^2 \mathrm{Var}(X) + \alpha^2 \mathrm{Var}(Y) + 2\alpha(1-\alpha)\mathrm{Cov}(X, Y). \quad (77)$$

*Cross term.* Assume noises are independent across clusters; then $X$ is independent of $Y$ and $\mathrm{Cov}(X, Y) = 0$.

*Bounding each variance.* From $\mathrm{Cov}(\varepsilon_c) \preceq \sigma^2 I$ and $\|\mathbf{e}\| = 1$,

$$\mathrm{Var}(X) = \mathrm{Var}(\langle \varepsilon_c, \mathbf{e}\rangle) = \mathbf{e}^\top \mathrm{Cov}(\varepsilon_c)\mathbf{e} \leq \sigma^2. \quad (78)$$

Independence across different $u$ yields

$$\mathrm{Var}(Y) = \mathrm{Var}\Big(\sum_u w_{cu}\langle \varepsilon_u, \mathbf{e}\rangle\Big) = \sum_u w_{cu}^2 \mathrm{Var}(\langle \varepsilon_u, \mathbf{e}\rangle) \leq \sigma^2 \sum_u w_{cu}^2 \leq \sigma^2 \sum_u w_{cu} = \sigma^2, \quad (79)$$

where we used $0 \leq w_{cu} \leq 1$ and $\sum_u w_{cu} = 1$.

*Combine.* Plugging the bounds into the decomposition (with $\mathrm{Cov}(X, Y) = 0$) gives

$$\mathrm{Var}\big(\langle \tilde{z}_c - \mu_c, \mathbf{e}\rangle\big) \leq \big((1-\alpha)^2 + \alpha^2\big)\sigma^2. \quad (80)$$

Finally, if neighboring means are similar along $v$, the last (bias) term is negligible, and $((1-\alpha)^2 + \alpha^2) = 1 - 2\alpha + 2\alpha^2 < 1$ for $\alpha \in (0,1)$, which gives the stated reduction. $\square$

# F   ORTHOGONALITY REGULARIZATION $\lambda_{\mathrm{ORTH}}$ (EQN. 15).

Table 8 shows the ablation of the trade-off parameter $\lambda_{\mathrm{orth}}$, which controls the strength of the orthogonality regularization. We can see that SimPlex-GT is not highly sensitive to this hyperparameter, but when compared with the variant without it (first row), the results show that orthogonal regularization still improves performance. This improvement can be attributed to its ability to reduce potential representation redundancy and interference, as discussed in Sec. 2.

Table 8: The effects of $\lambda_{\mathrm{orth}}$

| $\lambda_{\mathbf{orth}}$ | **Texas** | **Roman** | **Cora** |
|---|---|---|---|
| 0 | 91.89±4.05 | 80.23±0.55 | 83.50±0.11 |
| 0.3 | 92.54±3.66 | 81.12±0.53 | **84.05**±0.15 |
| 0.5 | **92.97**±4.39 | **81.56**±0.68 | 83.80±0.19 |
| 0.8 | 92.56±4.06 | 80.98±0.60 | 83.80±0.15 |

Table 9: Dataset statistics. "Homo" denotes the homophily ratio.

| Dataset | Nodes | Edges | Features | Classes | Homo |
|---|---|---|---|---|---|
| Cornell | 183 | 295 | 1,703 | 5 | 0.30 |
| Texas | 183 | 309 | 1,703 | 5 | 0.11 |
| Wisconsin | 251 | 499 | 1,703 | 5 | 0.21 |
| Actor | 7,600 | 29,926 | 932 | 5 | 0.22 |
| Chameleon (Filtered) | 890 | 17,708 | 2,325 | 5 | 0.24 |
| Squirrel (Filtered) | 2,223 | 93,996 | 2,089 | 5 | 0.21 |
| Roman-Empire | 22,662 | 32,927 | 300 | 18 | 0.05 |
| Cora | 2,708 | 10,556 | 1,433 | 7 | 0.81 |
| CiteSeer | 3,327 | 9,104 | 3,703 | 6 | 0.74 |
| PubMed | 19,717 | 88,648 | 500 | 3 | 0.80 |
| Ogbn-Arxiv | 169,343 | 1,166,243 | 128 | 40 | 0.66 |

## G  DATASET STATISTICS

We summarize the details of the datasets used in our experiments in Table 9.

## H  HYPERPARAMETERS

We provide the major hyperparameter search space:

- $\lambda_{\text{orth}}$: $\{0,\ 0.1,\ 0.3,\ 0.5,\ 0.8\}$.
- Learning rate for Encoder: $\{0.01,\ 0.005,\ 0.001\}$.
- Dropout for GCN: $\{0.1,\ 0.3,\ 0.5,\ 0.7,\ 0.8\}$.
- Dropout for Attention: $\{0.1,\ 0.3,\ 0.5,\ 0.7,\ 0.8\}$.
- Dimension of tokens: $\{128,\ 256,\ 512,\ 1024,\ 2048,\ 4096\}$.
- Total masking ratio: $\{0.9,\ 0.8,\ 0.5,\ 0.3,\ 0.1,\ 0\}$.
- Dynamic masking ratio: $\{0.9,\ 0.8,\ 0.5,\ 0.3,\ 0.1,\ 0\}$.
- Momentum: $\{0.9,\ 0.99,\ 0.999\}$.

## I  THE USE OF LARGE LANGUAGE MODELS (LLMS)

Large Language Models (LLMs) are used solely for polishing the writing.

