# OpenReview forum: "SimPlex-GT: A Simple Node-to-Cluster Graph Transformer for synergizing homophily and heterophily in Complex Graphs"
_ICLR.cc/2026/Conference — Submitted to ICLR 2026_

### Official Review · Reviewer_msha · 2025-10-30

**Soundness:** 3
**Presentation:** 2
**Contribution:** 2
**Rating:** 4
**Confidence:** 4

**Summary:**

SimPlex-GT is Graph Transformer designed to synergize homophilic and heterophilic patterns in complex graphs. It integrates local GCN-based message passing with a sparse node-to-cluster (N2C) attention mechanism, fuses local/global features via complementary filtering or cluster smoothing, and adopts a self-supervised teacher–student framework with dynamic masking. Evaluated on 11 benchmarks, it achieves state-of-the-art performance on heterophilic graphs, remains competitive on homophilic ones, and offers superior computational efficiency.

**Strengths:**

1.Effectively handles both homophily (via GCN) and heterophily (via N2C attention) without specializing in either.

2.N2C attention reduces complexity from O(N^2) to near-linear, enabling scalability for large graphs.

3.Outperforms existing GNNs and GTs on heterophilic datasets (e.g., Texas, Chameleon) while matching top results on homophilic ones (e.g., Cora, PubMed).

**Weaknesses:**

1.The performance of the proposed model is slightly sensitive to the number of clusters. How should this parameter be tuned for optimal results?

2.From the experimental results, most Graph Transformers perform worse than traditional GNNs. Does this suggest that designing on GTs is not meaningful? Meanwhile, traditional GNNs already have extensive work addressing both homophilic and heterophilic graphs. This gives the impression that improving GTs for handling homophily and heterophily may be unnecessary.

3.Figure 2 does not seem to convey much information.

4. In Equation 14, what does LN represent? Using $\mathcal{G}$ to denote the representation here can be misleading.

**Questions:**

See weaknesses.

---

> ### Author Response · Authors · 2025-11-27
>
> Thank you for your comments. We would like to provide the following response to address your concerns.
>
> ## Response to Weakness 1
>
> We appreciate the reviewer’s comment. As shown in Table 6, within a broad range $M \in \{2, 5, 10, 15\}$, the performance variation is relatively small (typically within
> $\approx 1$--$2$ accuracy points), indicating that SimPlex-GT is generally robust to this parameter across datasets.
>
> In practice, we tune $M$ on the validation set using a small discrete grid search. Intuitively, small datasets usually do not require a large number of clusters, otherwise, the model may degenerate into an N2N-style attention mechanism (also see Remark 1 for the theoretical intuition). For larger datasets, we use a slightly larger $M$. Concretely, we tune  $M \in \{2, 5, 10, 15\}$ as the default search space for all datasets and select the best value based on validation accuracy. We observe that
> (i) a small number of clusters (e.g., $M=5$) already yields strong performance on most datasets; and
> (ii) larger $M$ does not necessarily bring performance gains, while it increases computational cost linearly in $M$.
>
> Therefore, depending on the dataset size, performing a lightweight grid search over this small set on a validation dataset is an efficient way to obtain the optimal value.
>
> ## Response to Weakness 2
>
> The reviewer correctly observes that many existing Graph Transformers indeed underperform traditional GNNs. However, this is precisely the motivation behind our work, rather than evidence against the viability of GT research. As discussed in Section 2.1.1, most existing GTs are structure-agnostic—they overlook graph topology or rely solely on positional encodings to approximate structural information. The contribution of SimPlex-GT is to address exactly this limitation: we show that once the proper local inductive biases (via N2C, CF/CS, and the GNN residual branch) are introduced, Graph Transformers can consistently outperform traditional GNNs. In other words, our results demonstrate that the performance gap stems from missing structural priors, not from any intrinsic limitation of GTs.
>
> Moreover, traditional GNNs also have drawbacks: they lack the global inductive bias that Graph Transformers naturally provide. As a result, although GNNs perform strongly on homophilic graphs, they typically struggle on heterophilic graphs due to their inherently local message-passing nature (as shown in Fig. 2). While some recent methods attempt to address this issue, most of them either (i) cannot achieve strong performance simultaneously on both homophilic and heterophilic graphs (e.g., H2GCN), or (ii) rely on complex and specialized designs, such as contrastive learning between homophilic and heterophilic views in GREET. This further highlights the efficiency advantage of our model, which achieves strong performance on both regimes using a much simpler architecture. Furthermore, to capture long-range dependencies, GNNs usually need stack many layers, which leads to well-known problems such as over-smoothing, over-squashing, and scalability concerns. These issues highlight why incorporating global inductive bias without sacrificing efficiency is essential, and this is exactly what SimPlex-GT provides.
>
> Our model can be viewed as a principled combination of Graph Transformers and GNNs, capturing both local structural information and global feature dependencies, while still maintaining favorable computational complexity.
>
> ## Response to Weakness 3
>
> The purpose of Fig. 2 is to provide a preliminary study demonstrating the limitations of classical GNNs and the advantages of Graph Transformers in heterophilic settings. This figure offers a clear motivation for choosing a GT-based architecture as our backbone. Moreover, Fig. 2 empirically validates Theorem 1, which theoretically characterizes the failure modes of GNNs under heterophily.
>
> Moreover, Fig. 2 and Table 1 tell a complete and consistent story: Fig. 2 shows classical GTs excel on heterophilic graphs; Table 1 shows classical GTs struggle on homophilic graphs. Our method achieves the best of both worlds—strong performance on both homophilic and heterophilic graphs.
>
> ## Response to Weakness 4
>
>  "LN" represents layer norm. Specifically, in Eq. (14), we combine the two branches by directly adding them together. The combined representation is then passed through an MLP followed by a residual connection, and finally normalized with a layer norm (LN). We will make the notation more understandable in our final revision.

---

### Official Review · Reviewer_jBL4 · 2025-11-01

**Soundness:** 3
**Presentation:** 2
**Contribution:** 3
**Rating:** 6
**Confidence:** 3

**Summary:**

Traditional Graph Neural Networks (GNNs) struggle to handle graph structures under mixed patterns due to their inherent smoothing operations. To address this, the paper proposes SimPlex-GT, a novel graph transformer model. This model utilizes traditional local Graph Convolutional Networks (GCNs) for message passing to tackle homogeneity issues. Simultaneously, it designs complementary filtering and clustering smoothing mechanisms, and on this basis, constructs enhanced node attention mechanisms as well as global node-to-cluster attention mechanisms to deal with heterogeneity problems. SimPlex-GT can effectively process complex structural patterns.
In terms of training methods, SimPlex-GT adopts a self-supervised learning framework, with masked node modeling serving as the primary proxy objective, and employs a teacher-student prediction architecture. Additionally, it introduces a node-difficulty-driven dynamic masking strategy. This strategy can adaptively adjust the masking process, enabling the model to learn more robust and information-rich representations.
Through comprehensive theoretical analysis and outstanding empirical performance in terms of efficiency, the authors demonstrate that SimPlex-GT maintains a high level of competitiveness on homogeneous graphs in benchmark datasets, while also improving memory and training efficiency.

**Strengths:**

Overall, this article demonstrates strong innovation by proposing novel designs: complementary filtering and clustering smoothing. In complementary filtering, low-frequency information serves as the prototype, while high-frequency signals act as effective queries. By introducing graph structure awareness into the input (achieved through complementary filtering) or output (accomplished through clustering smoothing) of the node-to-cluster module, this design leverages the optimal characteristics of the two proposed models to collaboratively handle homogeneity and heterogeneity issues.
The local feature model focuses on neighborhood similarity, whereas the global feature model centers on dynamically learned clustering prototypes. A learnable gating mechanism integrates these complementary perspectives, and orthogonality constraints encourage diversity in representations. This design utilizes the collaborative modeling capabilities of the two branches to generate more robust node representations. It adopts masked node modeling as the primary proxy objective and employs a teacher-student prediction architecture.
Additionally, the paper balances task difficulty and data diversity by designing difficulty levels, maintaining a base masking rate, and preventing biased sampling.

**Weaknesses:**

When designing the two models to synergize homogeneity and heterogeneity, the authors took model simplicity as the starting point and opted for single-layer Graph Convolutional Networks (GCNs). Occasionally, the authors also mentioned that stacking multiple modules could enhance the model's performance. Therefore, it seems that improving model performance was not the primary consideration in model construction. Moreover, some variables or operations in the formulas, such as those in the gating mechanism, do not appear to have detailed explanations regarding their specific solutions or implementations.

**Questions:**

1）The paper employs single-layer GCNs in multiple places. For instance, 1) When designing complementary filtering, the author divides nodes into two complementary channels through a low-pass filter, which is implemented and approximated by a single-layer GCN. Can such a simple single-layer filter achieve the desired effect? 2） When using the graph structure to address homogeneity issues, a single-layer GCN branch is introduced as a residual path to capture this information and is integrated into the heterogeneous target N2C output. Can this single-layer GCN effectively capture the information? Is the model too simplistic?
2）In lines 290 - 294, the author states, "Note that, similar to other baselines (Rampášek et al., 2022), above designs can be viewed as a building block in our framework, and multiple blocks can be easily stacked to enhance the model’s expressive power. In our experiments, we retain a single block for simplicity." The author mentions that multiple blocks can be easily stacked to enhance the model's expressive power. Is it inappropriate to forgo enhancing the model's expressive power for the sake of simplicity?
3）The author mentions in the node-to-cluster (N2C) attention model the concepts of meaningful attention joint clustering and compact attention joint clustering. How can we determine whether an attention is meaningful and compact?
4）The teacher's output S(v;φ) and the student's output T(v;ψ) are not provided in the paper.
5）The author refers to it as "smart cosine similarity," but from the formula, it just seems to be an ordinary cosine similarity function with λorth as a balancing parameter. Where does the "smart" aspect come into play?
6）The paper repeatedly mentions that orthogonality regularization promotes representational diversity. For example, an auxiliary orthogonality regularization term is introduced to encourage the two branches to learn mutually enhancing features. Specifically, which entities are subject to the orthogonality constraints?
7）How is the gating mechanism implemented?

---

> ### Author Response · Authors · 2025-11-27
>
> Thank you for your comments and support. We would like to provide the following response to address your concerns.
>
> ## Response to Weakness 1
>
> We would like to clarify that the primary objective of this work is to **design simple yet efficient model that can simultaneously learn homophilic and heterophilic patterns on complex real-world graphs without sacrificing efficiency.** Our preliminary analysis shows that Graph Transformers are naturally well-suited for capturing heterophilic patterns, whereas traditional GNNs are typically not able to do so, but conventional Graph Transformers suffer from prohibitive quartic complexity, making them practically infeasible on large graphs (as evidenced in Table 1). This motivates our proposed Node-to-Cluster (N2C) attention with CS/CF, which retains the expressiveness of global attention while offering near-linear efficiency.
>
> Regarding model depth: we intentionally adopt a single block for two reasons.
> (1) Effectiveness: even with only one block, our model already achieves substantial improvements over strong baselines, demonstrating that the proposed attention mechanism is the key contributor.
> (2) Efficiency–expressiveness tradeoff: In our experiments, we found that adding more blocks yields only marginal performance improvements while introducing substantial additional computational overhead, which contradicts our paper’s goal of developing an efficient yet expressive model.
>
> In summary, the contribution of this paper is about improving performance on challenging mixed-pattern graphs while preserving (and even improving) computational efficiency.
>
> ## Response to Weakness 2 & Question 8 (gating mechanism)
>
> Thanks for your suggestion. We will add more explanations for several notations used in our revision. Specifically, for the gating mechanism in Eq. (14), as you mentioned, we combine the two branches by directly adding them together. The combined representation is then passed through an MLP followed by a residual connection, and finally normalized with a layer norm (LN).
>
> ## Response to Question 1
>
> We thank the reviewer for this insightful question. We would like to clarify that the number of layers is a hyperparameter that can be easily tuned. As discussed in the response to Weakness 1, our design goal is to maintain a balance between efficiency and performance. In our preliminary experiments, we observed that a single-layer GCN already provides strong performance under our setting, and increasing the number of layers doesn't lead to improvements while introducing additional computational overhead. Therefore, we chose to keep the architecture simple for our experiments. However, we provide a more detailed explanation below:
>
> In Appendix D (Lemma 1), we analyze a generic low-pass filter $G = U g(\Lambda) U^\top$ with $0 \le g(\lambda) \le 1$
> and show that the low-pass component $G x$ and its complementary high-pass residual $H x = (I - G) x$ are *nearly orthogonal* in the $\ell_2$ sense, i.e., their inner product is uniformly bounded by $\varepsilon \lVert x \rVert_F^2$. Achieving exact orthogonality would require $g(\lambda) \in \{0,1\}$ for all $\lambda$,  which is  different from continuous polynomial GCN filters.
> Stacking more GCN layers simply increases the polynomial degree and sharpens the low-pass behavior, but it does not turn $g(\lambda)$ into a strict $\{0,1\}$-valued projector; moreover, in practice it mainly leads to oversmoothing.
> Since our goal in CF is to obtain two complementary (approximately orthogonal) channels rather than exact spectral projectors, a single-layer GCN is sufficient and better aligned with our efficiency and stability goals.

---

> ### Author Response · Authors · 2025-11-27
> **Response — Page 2**
>
> ## Response to Question 2
>
> We appreciate this related question. First, as stated earlier, our design goal is to maintain a strong balance between performance and efficiency. Increasing the number of layers does not provide meaningful performance gains in our experiments, yet introduces clear computational overhead. Second, the local structural information in our model is not captured solely by the residual GCN branch; it is also effectively modeled by the CF/CS components (e.g., cluster structures in CS).
>
> Moreover, on graphs with heterophily or mixed homophily–heterophily structures, stacking multiple layers of standard message-passing GNNs (e.g., GCN-style mean aggregation) is widely known to amplify low-pass smoothing while suppressing high-frequency / heterophilic signals, which leads to over-smoothing. This would (i) weaken the discriminative heterophilic information that N2C is specifically designed to capture, and (ii) increase computational cost—both of which undermine the core advantages of SimPlex-GT (efficiency and stability).
>
> In summery, the GCN branch serves only to inject a lightweight local low-pass prior (a homophily bias) into the SimPlex-GT block. Simply deepening the GCN branch neither significantly improves the modeling of mixed structural patterns nor complements N2C; instead, it would further wash out heterophilic signals and unnecessarily increase computation.
>
> ## Response to Question 3 (our statement in lines 290 - 294)
>
> Thanks for pointing out this point. In fact, this is precisely the motivation behind our design. Although many baselines stack multiple blocks to enhance expressive power [1,2,3], they do so at the cost of efficiency, which contradicts our goal. We explicitly state that the performance of our model can be further improved by stacking more layers—this is consistent with common claims in prior works. However, our objective is to design a simple yet efficient architecture, so we intentionally use only one block to maintain the balance between expressiveness and computational efficiency, rather than “forgoing expressive power for the sake of simplicity.”
>
> Importantly, our results in Tables 2 and 3 demonstrate that even with a single block, our model outperforms baselines that use multiple blocks, such as SGFormer and GraphGPS. This empirically confirms that the effectiveness of SimPlex-GT comes from the proposed N2C mechanism itself rather than from increasing depth.
>
> [1] Do Transformers Really Perform Bad for Graph Representation?
>
> [2] SGFormer: Simplifying and Empowering Transformers for Large-Graph Representations
>
> [3] Recipe for a General, Powerful, Scalable Graph Transformer

---

> ### Author Response · Authors · 2025-11-27
> **Response — Page 3**
>
> ## Response to Question 4 (meaningful and compact)
>
> In Lines 162–163, our intention is to highlight a key difference from traditional graph clustering methods. Prior works typically rely on a fixed pre-processing clustering algorithm (e.g., METIS), which is not learnable and does not adapt to the downstream objective. In contrast, our method introduces a learnable clustering module that is optimized by the downstream SSL task. As a result, the attention mechanism becomes more meaningful because both the keys and values are computed from the learned clusters, rather than from a static partition.
>
> Moreover, the resulting clusters are more compact and semantically coherent, since the clustering process is directly guided by the N2C attention. This allows the model to capture local–global structural patterns more effectively than fixed clustering approaches.
>
> **(1) ``Meaningful attention'' means hierarchical structures awareness.**
> We consider attention to be meaningful when it respects and reflects the intrinsic hierarchical structure of the graph. As discussed in Section 2.1.1, unlike N2N attention which performs “interaction-before-aggregation” and therefore cannot effectively learn hierarchical structures (and often relies heavily on positional encodings), our N2C design enforces “aggregation-before-interaction” through a learnable clustering module. This design ensures that the attention mechanism operates on semantically aggregated prototypes rather than raw individual nodes, enabling it to capture richer structural semantics.
>
> **(2) ``Compact clustering'' means concentrated assignments.**
>
> Remark 1 further shows that the factor $r_c$ can be written in terms of
> the soft assignments $p_{i,c}$ as
> $$r_c=\frac{\sum_i p_{i,c}^2}{\Big(\sum_i p_{i,c}\Big)^2}=\sum_i w_{i,c}^2, \qquad w_{i,c} := \frac{p_{i,c}}{\sum_j p_{j,c}},$$
> and the effective number of nodes assigned to cluster $c$ is
> $$ n_{\mathrm{eff}}(c):=\frac{1}{\sum_i w_{i,c}^2}.$$
> Hence,
> $$
> r_c = \frac{1}{n_{\mathrm{eff}}(c)}.
> $$
> When the assignments are compact---i.e., nodes with similar features/labels concentrate on a small number of coherent clusters---$n_{\mathrm{eff}}(c)$ is large and $r_c$ becomes small, which directly tightens the variance bound above. Therefore, ``attention-for-compact-clustering'' refers to this joint effect: because the prototypes $p_c$ are used as keys/values in attention and are optimized end-to-end by the downstream loss, the model is encouraged to form clusters with (i) low within-cluster dispersion and (ii) sufficiently large effective size (small $r_c$), which in turn yields a better node to cluster attention.
>
> ## Response to Question 5 (teacher and student model)
>
> We will clarify this in the revision as follows. The teacher and student models share exactly the same architecture, as defined in Sections 2.1 and 2.2. The teacher model operates on the full graph, while the student model processes the masked graph. The teacher model does not require gradient computation; instead, its parameters are updated via the exponential moving average (EMA) of the student model’s parameters.
>
> Let $v$ denote the raw node embedding.
> The teacher network maps $v$ to a
> $C$-dimensional logit (or probability) vector:
> $S(v;\phi) \in \mathbb{R}^C,$
> and the student network produces its prediction
> $T(v;\psi) \in \mathbb{R}^C.$
> Our distillation loss is then computed between $S(v;\phi)$ and $T(v;\psi)$ as shown in Equ. (15).
>
> ## Response to Question 6
>
> We believe that we don't include the "smart" in our submission. Please feel free to let us know where you find it.
>
> ## Response to Question 7 (orthogonality regularization)
>
> The orthogonality constraints are specifically applied to the output feature representations of the two complementary branches before the final fusion stage.
>
> Concretely, as shown in Eq.~(15), the self-supervised loss is
>
> $\mathcal{L}(\phi)= \frac{1}{N} \sum_{v \in V}(\| S(v;\phi) - T(v;\psi) \|_2^2+ \lambda \cdot \mathrm{Cos}( f^{homo}(v),\, f^{hetero}(v) )),$
>
> where $f^{homo}(v), f^{hetero}(v) \in \mathbb{R}^d$ denote the
> ooutput representations from the N2C branch and the
> GNN branch, respectively, in the student model. Thus, the orthogonality regularization is applied **node-wise** to the pair $(f^{homo}(v), f^{hetero}(v))$ for all $v \in V$,
> encouraging the two branches to produce feature vectors that are nearly orthogonal in the representation space and therefore capture complementary information instead of overlapping.

---

### Official Review · Reviewer_XN1c · 2025-11-01

**Soundness:** 3
**Presentation:** 3
**Contribution:** 2
**Rating:** 4
**Confidence:** 3

**Summary:**

This paper presents a Graph Transformer architecture called SimPlex-GT, which aims to provide a unified representation learning framework for complex graphs with both homophilic and heterophilic properties. The core contributions of the paper are threefold: 1) A scalable, linear complexity "Node-to-Cluster (N2C)" attention mechanism that approximates global attention by allowing nodes to focus on a set of dynamically learned cluster prototypes; 2) Two theoretically motivated synergy mechanisms (Complementary Filtering (CF) and Cluster Smoothing (CS)) to fuse local information from the GCN branch with global information from N2C attention; 3) A novel self-supervised learning paradigm that employs a difficulty-driven dynamic masking strategy under a teacher-student framework. The authors demonstrate the effectiveness and efficiency of this approach through theoretical analysis and extensive experiments on 11 benchmark datasets.

**Strengths:**

1.Overall, the presentations are clear and easy to understand their framework and results.
2.For each core design (N2C, CF, CS), the paper provides corresponding theoretical support (Theorems 1-4), increasing the credibility of the method and clearly explaining its underlying working principles (e.g., the variance reduction property of N2C).
3.The experimental section is well-structured, with comprehensive benchmarking on multiple datasets. The detailed ablation studies (Tables 4, 5, 6, 7, 8) systematically validate the necessity and effectiveness of each component, making the experimental conclusions reliable.

**Weaknesses:**

1.The SOTA performance achieved by the model is the result of both the novel architecture (N2C+CF/CS) and its powerful self-supervised training strategy (dynamic masking). The ablation study (Table 5) does not address the collinearity between the training strategy and N2C in terms of their impact on performance.
2.In the methods section, the authors position the SimPlex-GT module as a general building block that can be "easily stacked to enhance model expressiveness" (Page 6, Lines 289-291). However, in the experimental section, they explicitly state, "For simplicity, we only kept a single module." Given the well-known depth bottleneck in the Graph Transformer field (i.e., stacking multiple layers can lead to performance degradation), validating this claim in a single-layer setup weakens the experimental support for the "stackability" assertion.
3.One of the motivations for introducing N2C attention is the potential use of "hierarchical node structures" within the graph (Page 3, Lines 160-161). This is a strong entry point, but in the subsequent theoretical and experimental analysis, the authors do not revisit this idea. The theoretical analysis primarily focuses on variance reduction rather than hierarchical representation capability (Theorems 2 and 4). As a result, the initial motivation does not fully close the loop in the final analysis.

**Questions:**

See weakness

---

> ### Author Response · Authors · 2025-11-27
>
> Thank you for your comments. We would like to provide the following response to address your concerns.
>
> ## Response to Weakness 1
>
> We thank the reviewer for raising this point. However, we think the term “collinearity” is not strictly applicable here, since the node-to-cluster (N2C+CF/CS) architecture and the self-supervised dynamic masking strategy are distinct sources of gain, and removing either one leads to a significant drop in performance.
>
> In Table 5, we completely disable each individual component to isolate its effect.
> Specifically, we (i) replace the self-supervised learning objective with a semi-supervised task, and (ii) replace the node-to-cluster (N2C) aggregation with a naïve node-to-node (N2N) variant.
> These controlled ablations allow us to directly measure the standalone contribution of the training strategy and the architectural design, respectively. Since the removal of either component results in performance degradation, the two techniques are complementary and synergistic, rather than "collinear". The SOTA performance is achieved not because one masks the ineffectiveness of the other, but because they address different challenges: N2C handles complex structural aggregation in mixed-pattern graphs, while the proposed SSL framework provides a robust learning objective.
>
>
> ## Response to Weakness 2
>
> We respectfully disagree with the reviewer’s statement that there is "a well-known depth bottleneck in the Graph Transformer field (i.e., stacking multiple layers leads to performance degradation)".
>
> We suspect the reviewer may be referring to depth-related failures such as over-smoothing and over-squashing, which are indeed well known limitations of traditional message-passing GNNs. However, these phenomena do not generalize to pure Graph Transformer architectures. In fact, several representative Graph Transformer papers explicitly report the opposite trend. For example, Graphormer[1] employs up to 12 layers and states that it “does not encounter the problem of over-smoothing.” Similarly, SGFormer[2] emphasizes that “existing models typically adopt a default design of stacking deep multi-head attention layers for competitive performance.” Other classical Graph Transformer frameworks such as GraphGPS[3] also integrate multiple attention layers to enhance expressivity. These findings collectively suggest that Graph Transformers are usually structurally immune to the depth-induced degradation observed in MPNN-style GNNs. Our claim is therefore fully consistent with prior work and aligned with practical experimental observations.
> If the reviewer is aware of any empirical evidence showing the opposite behavior for Graph Transformers, we would appreciate being pointed to it.
>
> Moreover, prior works also acknowledge the computational overhead associated with stacking many Transformer layers, which is also consistent with our discussion. Although our method enjoys near-linear complexity, we found that a single N2C layer already achieves strong performance, and therefore we keep a one-layer configuration for simplicity and efficiency in the main experiments.
>
> We will clarify those writing in revision. In summery: By "easily stacked to enhance model expressiveness", it means that as a building block our proposed SimPlex-GT module can be architecturally configured to form a deeper model with more learnable parameters to increase the model capacity and expressiveness when needed. By "For simplicity, we only kept a single module.", it only states the empirical design choices made for experiments in this paper based on the strong performance we obtained.
>
> [1] Do Transformers Really Perform Bad for Graph Representation?
>
> [2] SGFormer: Simplifying and Empowering Transformers for Large-Graph Representations
>
> [3] Recipe for a General, Powerful, Scalable Graph Transformer

---

> ### Author Response · Authors · 2025-11-27
> **Response — Page 2**
>
> ## Response to Weakness 3
>
> We respectfully point out that the "hierarchical structure" is not merely a motivation but is explicitly instantiated via our Node-to-Cluster (N2C) module, and Theorem 2 and 4 serve as the theoretical justification for the effectiveness of this hierarchical abstraction.
>
> First, **we want to emphasize that the theoretical advantages (such as variance reduction) arise directly from the hierarchical representation introduced by our design. Our theorems are designed to formally justify the benefits brought by hierarchical node structures.** For instance, the analysis in Theorem 4 is intentionally performed on the cluster representations within the coarse (cluster-level) graph. It proves that performing smoothing at this hierarchical level (rather than the node level) effectively reduces the directional variance of the prototypes.
>
>
> Second, we also provide empirical evidence that explicitly demonstrates the benefits of the proposed hierarchical structure. The CF/CS branch in SimPlex-GT directly operates on the node–cluster hierarchy. Moreover, our ablation results in Table 5 clearly isolate this effect: removing the N2C hierarchy and replacing it with a naïve N2N variant leads to a consistent degradation across multiple benchmarks, demonstrating the necessity of hierarchical aggregation. In Table 1, we further show the efficiency advantages of the proposed design. The propsoed hierarchical structure allows SimPlex-GT to achieve competitive or superior accuracy with significantly lower computational cost.
>
> Overall, both the theoretical analysis and the empirical evidence consistently support the central role of the hierarchy in enabling SimPlex-GT’s performance and efficiency.

---

### Official Review · Reviewer_vMFh · 2025-11-02

**Soundness:** 1
**Presentation:** 2
**Contribution:** 2
**Rating:** 2
**Confidence:** 4

**Summary:**

This paper proposes SimPlex-GT, a simple yet efficient graph Transformer framework that effectively synergizes homogeneous and heterogeneous structures in graphs by combining local GNN message passing with global node-cluster attention mechanisms. Trained under a self-supervised teacher-student framework and employing a dynamic masking strategy to focus on challenging nodes, the model achieves efficient, stable, and state-of-the-art node representation learning across diverse graph datasets.

**Strengths:**

The paper proposes a graph transformer model that attempts to address the challenges of heterogeneous graphs and reduce time complexity. Experimental results indicate that this approach appears to be effective.

**Weaknesses:**

The paper presents numerous theorems, but many of them are problematic.

1. The theoretical analysis in the paper primarily relies on an overly strong assumption: that node features can be directly aligned with labels. Introducing this assumption appears to detach the entire problem from its graph-based nature. Under this assumption, logistic regression becomes sufficient to address these issues.
2. Certain assumptions were not included in the theorem statement but were utilized in the proof. This renders the theorem either over-claimed or incorrect. For example, in the proof of Theorem 1, the author assumes that the activation of the GNN is linear and the weights are the identity matrix (Line 761), yet this assumption is not stated in the theorem itself. By the way, this assumption will make a GNN even weaker than a logistic regression.
3. In practice, GNNs can achieve classification performance on heterogeneous graphs that surpasses the bound proposed by the authors, thereby reducing the validity of their theorem.
4. Theorem 1 analyzes what appears to be GCNs rather than GNNs. Many simple GNNs can achieve performance comparable to ground truth under this theoretical framework, such as GAT.

Minors:
1. This method does not appear to be applicable to graphs that cannot be clustered.
2. No code was provided for reviewers to examine.

Given that the theoretical section constitutes a significant portion of the paper and the potential issues it may contain, I believe this paper is not ready for publication.

**Questions:**

See weakness.

---

> ### Author Response · Authors · 2025-11-27
>
> ## Potential misunderstanding of the main goal of our submission
>
> Thank you for your comments. With all due respect, we notice that your comments seems overly emphasize the theorems, especially Theorem 1, with potential misunderstanding of the main goal of our submission, as well as of details of Theorems. Before addressing your concerns in detail, we would like to clarify the main goal of this submission and the roles of introducing the theorems. **Our primary objective is to propose a simple and effective model that better synergizes homophilic and heterophilic patterns, rather than to introduce a fundamentally new theoretical framework.** The theoretical results we provide are intended to justify the effectiveness and efficiency of our design. Therefore, evaluating our submission almost-entirely by the comprehensiveness of the theorems would overlook the advantages of our novel model design and its demonstrated improvements in learning on complex graphs.
>
> ## Response to Weakness 1
>
> We respectfully disagree with this comment and believe there is a misunderstanding regarding the purpose of Theorem 1.
>
> First, as we clearly state, the “feature–label alignment’’ assumption in Theorem 1 is not intended to reflect real-world conditions. Instead, it is deliberately introduced to construct a controlled theoretical environment—a sanity check. **Our goal is to demonstrate that even under such an idealized setting, classical GNNs (based on local message passing) can still fail due to the structural noise introduced by heterophilic edges.**
>
> Specifically, under this clean-signal model, we rigorously show that a one-layer mean-aggregation GNN provably flips the sign of the predictive signal for heterophilic nodes ($\rho_i < 1/2$), whereas a Graph Transformer does not. This result highlights an inherent limitation of message-passing GNNs: their aggregation mechanism can corrupt the representations in heterophilic regions. In contrast, our node-to-cluster attention preserves this signal.
>
> Thus, rather than “departing from the nature of graphs,” our analysis reveals that GNNs blindly rely on local graph structure is harmful on heterophilic graphs.
>
> Second, regarding logistic regression: our experiment already includes MLP, which plays exactly the same role as LR. This observation actually supports our central argument: on strongly heterophilic graphs, ignoring graph structure (e.g., using MLP or LR) can outperform structure-based GNNs (see Table 2). However, such structure-agnostic models perform significantly worse than traditional GNNs and Graph Transformers on homophilic graphs, highlighting the need for a model—such as our proposed SimPlex-GT—that can adapt effectively to both regimes.
>
> However, real-world graphs are rarely purely heterophilic. Most are mixed and complex, and classical GNNs are known to perform better on locally homophilic regions. This motivates our proposal of SimPlex-GT, which successfully merges the advantages of them. Our experiments clearly show that SimPlex-GT significantly outperforms MLP, which in turn is stronger than LR.
>
> Finally, we want to emphasize that we do not assume feature–label alignment in our overall method or experiments. This assumption appears ONLY in Theorem 1.

---

> ### Author Response · Authors · 2025-11-27
> **Response — Page 2**
>
> ## Response to Weakness 2
>
> We appreciate the reviewer’s attention to the proof details. However, we respectfully point out a factual oversight in the review regarding the theorem statement, and clarify why the "GNN is weaker than LR" phenomenon is exactly what Theorem 1 intends to prove.
>
> First, the reviewer states that the assumption of linear activation "is not stated in the theorem itself." This is factually incorrect. Please refer to Theorem 1 (Page 3) in our paper, which explicitly states:
>
> "For a one-layer mean-aggregation (linearized) GNN without self-loop..."  Therefore, the theorem statement is precise and consistent with the proof.
>
> Importantly, this simplified GNN is deliberate and common: our goal in Theorem 1 is to isolate the effect of the graph aggregation operator under heterophily, not to characterize the full expressiveness of deep GNNs.
>
>
> Second, regarding comparison with LR, we argue that this "weakness" is precisely the theoretical insight we aim to reveal. In a heterophilic setting, the majority of a node's neighbors belong to different classes. A Logistic Regression model (or stronger MLP in our experiment) that relies solely on the node's own feature and ignores the graph would succeed perfectly under our assumption. However, a GNN forces an aggregation of neighbors. When isolating the structural effect, this aggregation "pollutes" the node's representation with opposing signals from neighbors. This is precisely the phenomenon we aim to expose: on heterophilic graphs, blindly applying message passing can make the model strictly worse than not using the graph at all.
>
> **Through this theorem, we aim to emphasize that the key reason for the degraded performance on heterophilic graphs lies in the message-passing mechanism itself—not in whether the feature transformation is the identity matrix or whether the activation function is linear. As shown clearly in Table 2, even with multiple layers of non-linear transformations, traditional GNNs still perform far worse than simple MLPs on strongly heterophilic datasets. This empirical evidence directly motivates our design: the problem lies in how information is aggregated (message passing), not in the expressiveness of the feature transformation alone.**
>
> Besides, the Graph Transformer block we propose is analyzed under the same assumption in our proof, and we prove that its attention structure can avoid this failure by effectively suppressing harmful heterophilic neighbors.
>
> Thus, We believe that theorem 1 is not about showing that “GNNs are more powerful than logistic regression,” but about rigorously contrasting different graph operators in a controlled variable setting. And all empirical evaluations do not rely on the linearize simplification; they are formulated for more general graph filters and attention mechanisms.
>
>
> ## Response to Weakness 3
>
> We respectfully disagree with the reviewer’s logic that superior empirical performance "reduces the validity" of our theorem. This comparison is scientifically unsound as it conflates a theoretical failure mode analysis with empirical system performance.
>
> First, Theorem 1 is *explicitly conditional*: it provides a performance bound for a common simplified setting, namely (i) node features that are perfectly aligned with labels, e.g. $x_i^{(0)} = y_i \mathbf{e}$, and (ii) a *one-layer linearized mean-aggregation GNN without self-loops and with $W^{(0)} = I$* as defined in Theorem 1. Under these assumptions, we prove that such a GNN necessarily misclassifies every locally heterophilic node with $\rho_i < 1/2$, and therefore its error rate is at least $|V_{\text{hetero}}|/|V|$. The corollary does not claim to provide a universal upper bound on the performance of arbitrary deep, nonlinear GNNs on arbitrary real-world datasets.
>
> Next, the purpose of Theorem 1 is not to assert that ``no GNN can ever outperform $|V_{\text{hetero}}|/|V|$'' on heterogeneous graphs. Rather, the theorem isolates the core aggregation mechanism to explain and motivate our design. Under a clean-signal toy model, we show that naive mean aggregation on heterophilic graphs can be systematically harmful, whereas the proposed Graph Transformer block avoids this issue. This insight holds precisely under the stated assumptions and is fully consistent with empirical observations: advanced architectures (including ours) achieve substantially better performance on real datasets, while classical GNNs typically fail in heterophilic settings. Thus, Theorem 1 provides a valid and necessary theoretical support for both our model design and our empirical findings.

---

> ### Author Response · Authors · 2025-11-27
> **Response — Page 3**
>
> ## Response to Weakness 4
>
> We respectfully disagree that this observation undermines the value or validity of our theoretical analysis.
>
> First, we want to emphasize that mean aggregation is a standard proxy for analyzing GNNs. It is common theoretical practice (e.g., SGC , GCNII) to use mean aggregation—the core mechanism of GCN—as the representative proxy for the message-passing family.
>
> Our theorem targets the issue of message passing. By proving that mean aggregation fails in heterophilic settings, we expose the fundamental flaw of the it: forcing nodes to mix with neighbors is harmful when neighbors are heterophilic. This conclusion applies broadly to the class of GNNs that rely on homophily assumptions.
>
> Second, GAT does not solve the heterophily problem (both theoretically and empirically).  Theoretically, GAT computes a convex weighted combination of neighbor representations instead of mean. For example, under a highly heterophilic region, most of neighbors provide a misleading signals. Even with attention, computing a weighted average of noisy vectors still produces a misleading vector. Thus, GAT fails for the same fundamental reason. Empirically, we explicitly evaluate GAT in our experiments (see Figure 2 and Table 1). And on highly heterophilic datasets (e.g., Texas, Cornell), GAT significantly underperforms SimPlex-GT.
>
> ## Response to Minor Weakness
>
> We want to emphasis that our method never assumes the existence of externally given or ground-truth clustering, nor does it require the graph to exhibit clear community structure in the classical sense. The cluster assignments $P \in \mathbb{R}^{N \times M}$ in SimPlex-GT are **learned soft assignments, which fundamentally differentiates our approach from prior clustering-based baselines**. Formally, each row of $P$ lies on the probability simplex (via a row-wise softmax); thus, for any finite graph and any $M \ge 1$, $P$ always defines a valid soft partition of the nodes into $M$ prototypes. In this sense, there is no notion of a finite graph that ``cannot be clustered'' in our framework: these prototypes are learned representations over which attention is computed, not hard communities required as a prerequisite.
>
> For graphs that are intrinsically difficult to cluster or exhibit little meaningful community structure, N2C attention naturally reduces toward a global-pooling-like form, yet this still constitutes an effective representation-learning operation and remains a valid and expressive architecture for such graphs.
>
> Furthermore, our empirical results span a wide range of benchmarks with varying homophily ratios. For example, on highly heterophilic datasets such as Roman-Empire, our N2C module does not rely on homophily-based communities, yet still demonstrates stronger performance than competitive baselines.
>
> Overall, our approach should be viewed as a flexible, learnable, prototype-based attention mechanism that **adapts** to the underlying graph structure, rather than a method that depends on fixed or pre-existing clusters. We would be happy to further address this concern if the reviewer can provide a concrete example of a graph that ``cannot be clustered''.
>
> About the code, we believe that we have already stated in the abstract that the "code will be released upon acceptance".

---

### Meta-Review · Area_Chair_wPAH · 2025-12-07

**Summary:**

Reviewers pointed out concerns about the rigor of theorems, ablation study, universality validation, and clarity. Some of them are partially answered, including clarity and an ablation study. However, the rigor of theorems remains weak. Therefore, I think this paper needs further polishing.

**Reviewer Concerns:**

Reviewer vMFh pointed out that most theorems are not rigorous,  and I think the authors’ responses did not comprehensively address these concerns after reading the rebuttal feedback. Reviewer msha thinks the stability of the performance improvement is weak, and I agree with him by checking the ablation study. Reviewer XN1c pointed out the lack of sufficient ablation study and validation on universality, which are partially answered.  Besides, some clarity concerns raised by multiple reviewers are properly answered.

**Reviewer Scores:**

Based on the unsolved rigorous theorem, reviewer vMFh may not change his scores. Taking into account the clarification of details， reviewers msha or XN1c may change their rating from 4 (weak reject) to 6 (weak accept). Thus, I think the final rating may be (6, 6, 4, 2).

---

### Decision · Program_Chairs · 2026-01-26

Reject